# Perspectives of health and community stakeholders on community-delivered models of malaria elimination in Lao People's Democratic Republic: A qualitative study

**May Chan Oo** [1][*], **Khampheng Phongluxa** [2], **Win Han Oo** [1,3][*],
**Sengchanh Kounnavong** [2], **Syda Xayyavong** [2], **Chanthaly Louangphaxay** [2], **Win Htike** [1],
**Julia C. Cuts** [3], **Kaung Myat Thu** [1], **Galau Naw Hkawng** [1], **Freya J. I. Fowkes** [3,4,5]

**1** Health Security Program, Burnet Institute Myanmar, Yangon, Myanmar, **2** Lao Tropical and Public Health Institute, Vientiane, Lao PDR, **3** Disease Elimination Program, Burnet Institute, Victoria, Australia, **4** Melbourne School of Population and Global Health, University of Melbourne, Victoria, Australia, **5** Department of Epidemiology and Preventive Medicine, Monash University, Victoria, Australia

☯ These authors contributed equally to this work.
* maychan.oo@burnet.edu.au (MCO); winhan.oo@burnet.edu.au (WHO)

**Data Availability Statement:** According to the National Ethics Committee for Health Research

## Abstract

In the Lao People's Democratic Republic (Lao PDR), village health volunteers play an important role in providing health services including those to reduce the burden of malaria. Over the last two decades, the volunteer network has expanded to bring malaria services closer to communities and contributed to the reduction of malaria cases. However, as malaria test positivity rates decreased, many volunteers have lost motivation to continue providing routine malaria services, and other services they provide may not reflect growing healthcare demands for common diseases in the community. This study explored the perspectives, knowledge and inputs of key health stakeholders and community members in southern Lao PDR on community-delivered models in order to refine the volunteer model in the context of Lao PDR's primary health care sector and malaria elimination goals. Semi-structured interviews with multi-level health stakeholders, participatory workshops with community leaders, and focus group discussions with community members and current village health volunteers were conducted. Deductive followed by inductive thematic analysis was used to explore and categorise stakeholders' perspectives on community-delivered models for malaria elimination. Both stakeholders and community members agreed that village health volunteers are essential providers of malaria services in rural communities. Apart from malaria, community members identified dengue, diarrhoea, influenza, skin infections and tuberculosis as priorities (in descending order of importance) and requested community-based primary health care for these diseases. Stakeholders and community members suggested integrating prevention, diagnosis, and treatment services for the five priority diseases into the current malaria volunteer model. A divergence was identified between community members' expectations of health services and the services currently provided by village health volunteers. Stakeholders proposed an integrated model of healthcare to meet the needs of the community and help to maintain volunteers' motivation and the long-term

(NECHR), Lao PDR) confidentiality must be maintained utmost. Therefore, the datasets generated and/or analysed during the current study are not publicly available as the study collected data from specific province, districts and villages in Lao PDR, and the information may be identifiable to particular individuals, risking a breach in confidentiality. Data access requests may instead be sent to: National Ethic Committee for Health Research; Dr. Phoutanong Thongmalayvong Email: nechr@gmail.com, Tel: +856-21-250670 ext 205, 209.

**Funding:** The funding was received from the National Health and Medical Research Council (Australian Centre for Research Excellence in Malaria Elimination (ACREME) (1134989) and Career Development Fellowship to FJIF (1166753); and its Independent Research Institute Infrastructure Support Scheme). The funders had no role in study design, data collection and analysis, decision to publish, or preparation of the manuscript.

**Competing interests:** The authors have declared that no competing interests exist.

**Abbreviations:** DHIS2, District Health Information System2; FGD, Focus group discussions; GMS, Greater Mekong Sub-region; IDI, In-Depth Interviews; KII, Key informant interview; Lao PDR, Lao People's Democratic Republic; MoH, Ministry of Health; MDA, Mass Drug Administration; RDT, Rapid Diagnostic Test; TB, Tuberculosis; UHC, Universal Health Coverage; VHV, Village Health Volunteer; VMW, Village Malaria Worker.

sustainability of the role. An evidence-based, integrated community-delivered model of healthcare should be developed to balance the needs of both community members and stakeholders, with consideration of available resources and current health policies in Lao PDR.

## Introduction

In 2020, nearly half of the world's population (approximately 3.4 billion people) was at risk of malaria, while an estimated 241 million malaria cases and 627,000 malaria-associated deaths were reported globally [1]. The Greater Mekong Sub-region (GMS), which comprises Cambodia, China (Yunnan Province), Lao People's Democratic Republic (Lao PDR), Myanmar, Thailand and Viet Nam [2] has seen dramatic declines in malaria burden over previous decades: between 2000 and 2019, annual reported malaria cases fell by 90% [3,4].

After acknowledging the threat of drug-resistant malaria [5], Asia Pacific leaders committed to malaria elimination in the GMS by 2030 [6]. In Lao PDR, this malaria elimination target may be feasible, given that reported national annual parasite incidence declined from 6.81 to 1.3 per 1,000 population between 2012 and 2018 [2]. In 2018, malaria transmission is low and sporadic in the northern provinces but higher in the southern provinces, which account for 95% of all malaria cases in the country [7]. The national strategic plan for malaria control and elimination 2016–2020 focuses on burden reduction in five southern provinces and targets elimination in 13 northern provinces [7,8].

In parallel with the switch from malaria control to elimination in GMS, the motivation and social role of malaria volunteers have quickly plummeted along with the decline of malaria burden [9]; therefore, the sustainability of the community-delivered model is impacted. Evidence has shown that in the context of lower malaria incidence, increasing the services that the community requested, and the malaria volunteers can provide, increases utilisation and enables more malaria testing [10]. For example, the integrated Community Case Management model that delivers case management in malaria, pneumonia, diarrhoea, malnutrition screening and nutrition counselling to children under five, has shown good acceptability in Africa [11,12] and proved its effectiveness in reducing under five mortality and malaria burden in Africa [13–17] as well as Myanmar [18].

In Lao PDR, community-delivered models are used to implement key malaria interventions such as testing and treatment and bed nets, which have contributed to the declining malaria burden over the past decade. Throughout Lao PDR, there is a large community-based network of village health volunteers (VHVs) and village malaria workers (VMWs) providing services in hard-to-reach areas [19,20]. Malaria program-specific VMWs are assigned to provide malaria services only in remote high malaria transmission areas (annual parasite incidence > 10 per 1000 population) in Lao PDR whereas VHVs provide the same malaria-specific services as well as other general integrated health services in hard-to-reach areas where there is no health centre [8,21,22]. The specific services provided by VHV and VMW are detailed in supporting file 1. Malaria services provided by both VMW and VHV include malaria testing and treatment, net distribution, education and surveillance, with VHV also providing general health services such as assisting health centre staff in outreach activities, providing health education and community health promotion services, offering basic first aid care, facilitating antenatal care and performing vital event surveillance (S1 File) [19,20]. VHV provide the vast majority of services across Lao PDR; in 2020, 92% of volunteers (13,089/14,227) were VHVs compared to 8% of volunteers (1,138/14227) who were VMWs [7,23].

The current packages of services provided by VHVs and VMWs was developed according to operational feasibility and the requirements of the Lao PDR Ministry of Health (MoH), without community consultation. Therefore, they may not reflect growing healthcare demands for common diseases in the community [20]. Moreover, as the malaria burden has declined in Lao PDR and the malaria test positivity rates performed by VHVs and VMWs have decreased, many VHVs and VMWs may have lost motivation to continue providing routine malaria control services in the community [7].

Lao PDR is transitioning from a malaria control to a malaria elimination program. In 2018, Distinct Surveillance and Response Guidelines were developed for case-based surveillance, focus investigations and responses in Lao PDR, whereby case notification is conducted within a day after diagnosis of a malaria case, individual case and foci investigation within three days and response within seven days (1-3-7 model). However, the 1-3-7 strategy has so far failed to be applied effectively without the help of VHVs and VMWs in Lao PDR [7]. Furthermore, it is doubtful that current community-delivered models of VHVs and VMWs meet the requirements of the malaria elimination program in Lao PDR, such as real-time notification of malaria cases within 24 hours, and case-based surveillance and response activities.

In this context, the current VHV and VMW model in Lao PDR needs to be reviewed and modified so it can support the malaria elimination program and primary health care most effectively. It should be an operational and pragmatic community-delivered malaria elimination model, acceptable to key health stakeholders and community members in the context of Lao PDR's primary health care sector. To generate information to enable development of a new model in line with the recommendations of key health stakeholders, VHVs and community members, a qualitative study was conducted in three districts of Savannakhet Province in Lao PDR in 2019.

## Materials and methods

### Study setting, data collection methods and participants

This study was conducted in Nong, Sepone and Atsaphone districts in Lao PDR's southern Savannakhet Province, whose area are 1,833, 2,097, 1,482 km$^2$ respectively. The population density of Nong district is 18.75/km$^2$ while Sepone and Atsaphone are 31.06/km$^2$ and 43.69/km$^2$ respectively. The majority of the population are farmers and live in mountainous and forested areas [24]. Among the three districts, Nong and Sepone are located close to the Vietnam border. These districts were purposively selected for inclusion based on the logistical feasibility of conducting research. When the study was conducted in mid-2019, Nong and Sepone were in the malaria burden reduction (control) phase and Atsaphone was in transition to the malaria elimination phase (Fig 1).

Qualitative methods of semi-structured interviews, focus group discussions (FGDs) and participatory workshops were used to investigate the perspectives, knowledge and inputs of key MoH stakeholders (ranging from provincial to field office levels), VHVs, community members, and community leaders on community-delivered models for malaria elimination in Lao PDR. Pre-tested topic guides or facilitation guides (S2 File) were used in all components of data collection. Participants were recruited purposively by in-country research team members. All interviews, FGDs and workshops conducted between May and June 2019 were undertaken in Laotian.

**Semi-structured interviews.** Key informant interviews were used to collect data from high-level policymakers, decision-makers and managers from MoH, Lao PDR and, given their generalized knowledge of community-delivered models and malaria. Key informant interviews are qualitative in-depth interviews with people who know what is going on in the community.

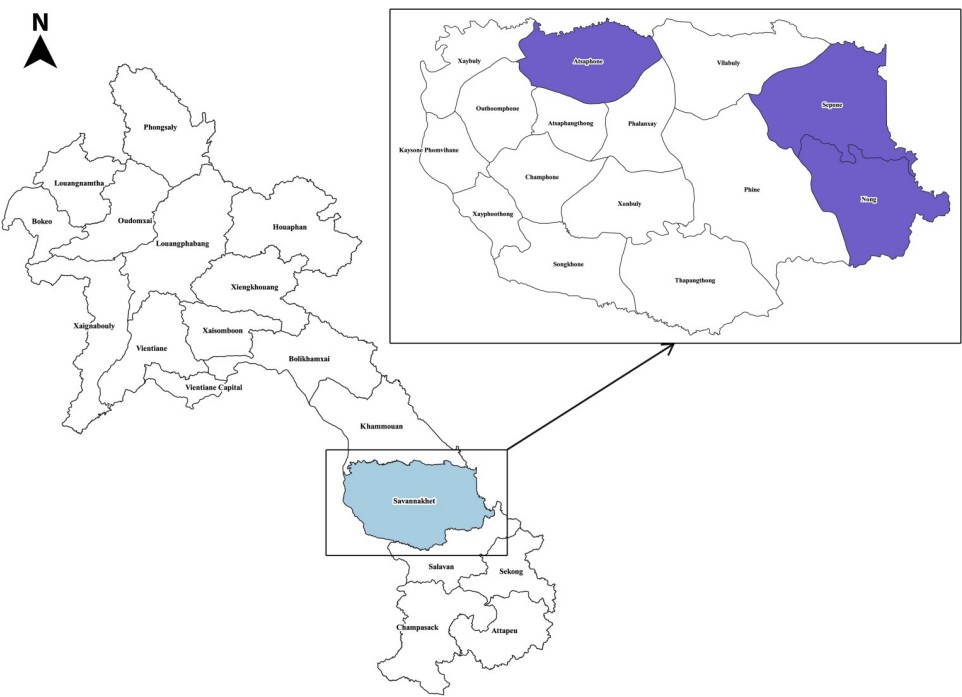

**Fig 1. Map of studied districts in Savannakhet province, Lao People's Democratic Republic (this figure is developed by the authors by using QGIS software (version 3.22.0) and Photoshop 2020.** The figure is licensed under a CC BY 4.0.).

The purpose of key informant interviews is to collect information from a wide range of people —including community leaders, professionals, or residents—who have first-hand knowledge about the community [25]. In-depth interviews were conducted with middle-level managers, staff and technical advisors from MoH, Lao PDR and malaria IPs, to take advantage of their knowledge and personal experiences in managing, implementation and operating the community-delivered models and malaria interventions [26,27]. In-depth interviewing is a qualitative research technique that involves conducting intensive individual interviews with a small number of respondents to explore their perspectives on a particular idea, program, or situation [28].

Key informant interviews (KII, n = 4) and in-depth interviews (IDI, n = 9) were conducted with MoH stakeholders who were purposively recruited based on their role in the ministry, interest in and experience with community-delivered models for malaria, and the operational feasibility including logistical arrangement of the interview (Table 1). The question guide of semi-structured interviews include six themes which are (i) current malaria situation and control measures in their respective working area (ii) views and perspectives on the current malaria VHV models (iii) policy and strategic barriers and enablers for malaria elimination (iv) operational barriers and enablers for malaria elimination using community delivered models (v) strategies to maintain the motivation and social role of VHVs in the community (vi) factors that need to be addressed during the transition from malaria control to elimination using community delivered models. Each interview was conducted in person by a trained interviewer or research team member from the Lao Tropical and Public Health Institute, in a private location. Interviews lasted between 45 and 60 minutes.

**Focus group discussions and participatory workshops.** In both workshops and FGDs, group discussions with open-ended questions were conducted to facilitate an interactive and

**Table 1. Socio-demographic characteristics of the study participants.**

| Characteristics | Group discussion | | Semi-structured interview | |
|---|---|---|---|---|
| | Focus group discussion n (%) | Workshop n (%) | In-depth interview n (%) | Key informant interview n (%) |
| **Number of participants in total** | 169 (100.0) | 75 (100.0) | 9 (100.0) | 4 (100.0) |
| **Position/designation** | | | | |
| 1. Villagers/village leaders/volunteers | 169 (100.0) | 75 (100.0) | - | - |
| 2. Health centre staff | - | - | 6 (66.7) | - |
| 3. District health staff/district health malaria unit staff | - | - | 3 (33.3) | 3 (75.0) |
| 4. Provincial Officer | - | - | - | 1 (25.0) |
| **Level of representation** | | | | |
| Community level | | | | |
| 1. Sepon | 54 (31.9) | 18 (24.0) | 2 (22.2) | - |
| 2. Nong | 62 (36.7) | 30 (40.0) | 2 (22.2) | - |
| 3. Atsaphone | 53 (31.4) | 27 (36.0) | 2 (22.2) | - |
| District level | | | | |
| 1. Sepon | - | - | 1 (11.1) | 1 (25.0) |
| 2. Nong | - | - | 1 (11.1) | 1 (25.0) |
| 3. Atsaphone | - | - | 1 (11.1) | 1 (25.0) |
| Provincial level | - | - | - | 1 (25.0) |

participatory atmosphere, allowing participants to express their opinions freely and interact with each other to build responses. The guides of FGDs and participatory workshop include seven themes which are (i) current malaria situation and priority health problems in the respective community (ii) malaria control measures and available health services in the respective community (iii) views and perspectives on the current malaria VHV models (iv) policy and strategic barriers and enablers for effective malaria control and elimination in the community (v) available community supports for malaria control and elimination in their community (vi) strategies to maintain the motivation and social role of VHVs in the community (vii) culture, customs and norms of the ethnic communities that paly as barriers and enablers for effective malaria control and elimination in the community. FGDs and workshops were conducted in secure and private locations such as temples, community leaders' houses or village offices. Whilst both VMW and VHV work closely together to provide malaria services and were invited to participate in the three studied districts, only VHVs participated in this study because during the survey collection period, VMWs were not available to participate in this study. Eighteen FGDs with VHVs and community members (combined but men and women separately; nine FGDs each) and nine participatory workshops with community leaders were undertaken (Table 1).

The FGDs varied in length from one to two hours. They were facilitated by an in-country research team member and supported by a research assistant serving as note taker/interpreter.

The participatory workshops were led by the in-country principal investigator assisted by at least two in-country research team members and averaged six hours. The research team wrote a detailed agenda, including the roles and responsibilities of facilitators (S2 File), prior to the workshops. In brainstorming sessions within the workshops, participants were grouped according to their occupation, geographical location and residential district in order to facilitate smooth discussions and effective outcomes. Multiple techniques, such as preference ranking of health services, matrix scoring, and social and resource mapping were used in the workshops to stimulate participation and to generate rich data [29,30].

## Data management and analysis

A qualitative descriptive approach was applied to explore the experiences, opinions and knowledge of stakeholders, and volunteers, community members and leaders–the beneficiaries of the model. This approach enabled a comprehensive summary of the participants' interactions and experiences with specific community-delivered models [31,32]. Interviews, FGDs and workshops were audio-recorded; the recordings were transcribed verbatim and translated into English for thematic analysis.

Deductive thematic analysis was applied initially, followed by inductive thematic analysis. This procedure was used because of the anticipated diversity of opinions and views of the participants. Data analysis was iterative throughout data collection because the two processes occurred concurrently, and included data immersion, coding, categorisation of sub-themes, and major theme development. There are basically three levels of qualitative data analysis namely micro, surface and macro levels, and the surface level analysis focus in themes, social patterns and understandings. In this study, the analysis was mainly conducted at a surface level [33], and explored patterns in the perspectives and experiences of the participants and generated new understandings. The first and second authors discussed the themes and sub-themes to reach the consensus. Triangulation of the data from semi-structured interviews, workshops and FGDs during analysis and reporting strengthened the validity of the findings. The key findings relate the themes explicitly to the aim of the study and the literature. Key findings are illustrated herein with examples from the data.

**Ethical considerations.**   The study protocol was reviewed and approved by: (1) the National Ethics Committee for Health Research, MoH, Lao PDR (Submission ID– 2019.47.sav) and (2) the Alfred Hospital Ethics Committee, Melbourne, Australia (Project Number—189/19).

Information outlining the focus of the study, the general conduct of the data collection, confidentiality and participants' roles was provided verbally and in a written document to all participants using Lao language. After participants had read and indicated their understanding of the information document, written informed consent was obtained.

## Results

### Community and stakeholders' perceptions on malaria burden

For the purpose of reporting, here provincial and district level health officers are referred to as high-level stakeholders and community-level health officers as middle-level stakeholders. VHVs and community members are collectively referred to as community members because VHVs also represented their views as members of the community. Most of the stakeholders reported that malaria prevalence in Savannakhet Province is declining, although malaria transmission remains high in some districts.

> *The malaria transmission in our province [Savannakhet] is steadily declining but there are still a number of high-risk and epidemic districts in this year [2019], such as Nong and Sepon.* (High-level stakeholder)

Stakeholders considered the volunteer service provision network, under the supervision of district malaria health units, to be the main contributor to the implementation of effective community-based malaria intervention activities and declining malaria prevalence. The scale-up of District Health Information System 2 (DHIS2) [34], which enables provincial and district health offices to respond to malaria case reports in a timely manner, technical and funding support from national and international organisations, and community participation in malaria prevention activities were also considered significant contributors to declining malaria cases.

Community members also recognised the decreasing malaria prevalence, but still considered the disease to be a major health problem. They perceived that malaria poses a huge burden on patients, their families and communities, harming their health, economic, education and social conditions.

## Priority health problems

Community members listed priority health problems in addition to malaria, namely dengue, diarrhoea, influenza, tuberculosis (TB), skin infections, conjunctivitis and non-communicable diseases such as diabetes, hypertension and gastritis. When these health problems were ranked according to morbidity, mortality, severity and contagiousness, dengue fever was rated the most dangerous and commonest health problem in the community (the three districts experienced a large and lethal dengue outbreak in 2017).

*Like malaria, most of us are also afraid of dengue fever. It is a common health problem in my village. Once a patient gets sick with dengue fever, there will be bleeding from the gum. Although dengue fever is treatable, it can change to a serious situation within two or three days. If treatment is not taken in time, the patient can soon die. So terrible.* (Community member, participatory workshop, Atsaphone District)

Community members nominated diarrhoea as the second most important health problem in the community. They reasoned that untreated diarrhoea can be fatal.

*We choose diarrhea because people can die within a short time if the treatment is not taken immediately"* (Community member, participatory workshop, Atsaphone District)

Nevertheless, stakeholders perceived that community members paid insufficient attention to environmental sanitation. Most of them drink surface water without boiling it first, which was believed to be a root cause of diarrhoea. Influenza, skin infection and TB were also considered significant health problems because of their infectivity and high prevalence in the community (Box 1).

## Health services in the community

Stakeholders reported various primary health care providers offering health promotion, prevention, diagnosis and treatment for communicable and non-communicable diseases–as well

### Box 1. Non-malaria priority health problems in the community

1. Dengue
2. Diarrhoea
3. Influenza (fever, sneezing and coughing)
4. Skin infection
5. Tuberculosis

as maternal and child health care–at the community level in Lao PDR. Volunteers equipped with rapid diagnostic tests (RDTs) and anti-malarial drugs for malaria services provision in malaria endemic villages are called VMWs. In non-malaria endemic villages, VHVs are equipped with first aid kits so they can provide basic health care services. International and local non-governmental organisations also provide health care services for both malaria and non-malarial illness in the communities studied (Box 2).

**Malaria health care services.**   If RDT tests are positive, VHVs provide malaria treatment according to Lao PDR National Malaria Treatment Guidelines, and palliative medicines like anti-pyretics. Most of the community members reported being satisfied with VHVs' malaria services. VHVs refer patients who are RDT-negative but feverish to health centres or hospitals for comprehensive care.

*When someone gets sick with malaria, the VHV gives anti-malaria drugs and other required treatments. So, the patient recovers from illness. For those patients with severe illness, the VHV provides a referral letter for a health centre or hospital.* (Community member, male FGD, Nong District)

Community members perceived that staff at health centres and hospitals provide more comprehensive care for severe fever cases than volunteers. However, in one FGD, participants discussed the transportation difficulties and costs associated with accessing health centres far from their villages. Because of these challenges, community members preferred to seek care

---

### Box 2. Available health care providers for malaria and non-malaria health problems in the community

Malaria health service providers

- Village health volunteers/ village malaria workers
- Village health centers
- District and provincial hospitals
- Local and international non-governmental organizations
- Spiritual healers

Non-malaria health service providers

- Village health volunteers
- Village health centers
- District and provincial hospitals
- Local and international non-governmental organizations
- Local pharmacies
- Private clinics
- Hospitals in Vietnam
- Spiritual healers

---

for suspected malaria initially from VHVs, who are easily accessible anytime in their villages, rather than health centres or hospitals.

In the FGDs, community members also identified that they received malaria health care services and awareness-raising activities from mobile medical teams of local and international non-governmental organisations. One international organisation trained retailers to sell anti-malarial drugs in remote villages to increase access to timely treatment.

Importantly, community members, leaders and VHVs did not understand the concept of subclinical malaria nor unique challenges of eliminating *Plasmodium falciparum* and *Plasmodium vivax* and did not discuss them in any FGDs or participatory workshop, despite prompts. In contrast to community members, leaders and volunteers, the stakeholders understood sub-clinical malaria however, they did not suggest any interventions for its diagnosis and treatment (such as the use of highly sensitive RDT [35] for diagnosis and mass drug administration (MDA) for elimination [36]). Unlike community members and leaders, the stakeholders concern the challenges of *P. vivax* elimination compared to *P. falciparum* elimination, but they did not recommend specific interventions for *P. vivax* elimination.

**Non-malaria health care services.** Apart from malaria health care services, community members perceived that VHVs do their best in providing basic treatment services using their first aid kits, assisting health centre staff, referring patients to health care facilities and facilitating antenatal care in the community. Community members choose service providers for their illness depending on the severity of the disease and easy access to the provider.

> *It depends on the disease. If the disease is severe, I normally go to the hospital directly. If it is not severe, I go to the VHV because the VHV has some medicine.* (Community member, female FGD, Atsaphone District)

Community members living in the border area villages reported receiving healthcare from a hospital in neighbouring Viet Nam as well as health facilities in Lao PDR. They were satisfied with the quality of care they received for the cost incurred.

> *Villagers go to the Viet Nam hospital because payment is cheaper than the district hospital and, medicine and equipment in the Vietnam hospital are complete.* (Community member, male FGD, Nong District).

Some community members seek health care from traditional healers. A few people reported using spiritual healing or seeking an exorcist for spells and used traditional medicine to heal their illness first; if it was not resolved, they went to conventional health care facilities.

> *We invite the shaman to treat our illness. We prepare the decoration and offer our livestock such as ducks, chickens, pigs and cows to our ancestors under the guidance of the shaman.* (Community member, participatory workshop, Nong district)

## Stakeholders' and community members' preferred community-delivered model

**Malaria elimination and disease integration.** Community members requested that current malaria interventions–such as distributing bed nets and mosquito repellents and conducting health education sessions–be continued. Stakeholders and community members agreed that a primary health care service package covering malaria and other health problems, provided by community-based volunteers, is needed.

*We want volunteers who are able to provide primary health care services not only for malaria but also for other common diseases in our village.* (Community member, participatory workshop, Nong district)

In their ideal package, community members identified services for the priority diseases of dengue fever, diarrhoea, influenza, skin infection and TB. Furthermore, the stakeholders recommended integrating the services for priority health problems identified by community members into the current VHV model.

To address these health priorities, community members want VHVs to lead awareness raising, environmental sanitation and pit latrine construction. Stakeholders also suggested that VHVs are effective at raising awareness about common health issues and improving health outcomes in the community. Community members suggested the health education methods they would like VHVs to deliver. They discussed videos and films were a popular choice among the younger generation, and health education messages through an audio-visual approach is suitable to persuade those age groups.

*We want to conduct health education sessions in the community by illustrating with pictures, videos and photos so that illiterate people or children could understand.* (Community member, participatory workshop, Adsaphone district).

Some female FGD participants requested health education sessions delivered by microphones and loudspeakers, because many Lao women cannot attend health education sessions because they are busy with household chores. If loudspeakers are used, they can listen whilst working.

Community members called for 24-hour, comprehensive health care services with home visits. They wanted volunteers to be able to give intramuscular and intravenous injections, because they believe that injections are better than oral treatment.

**Volunteer assignment and recruitment.** Stakeholders and community members generally agreed that one VHV per village is insufficient to combat malaria and other priority health problems. Community members suggested that having multiple volunteers in their community would enable them to choose a service provider depending on their level of trust, the type of illness and its severity, transportation cost and distance from the health facility. They requested at least three volunteers per village to tackle common diseases in the community.

*We would like to have three VHVs in the village because if one is absent, another one will be on duty so that it would be convenient for patients.* (Community member, participatory workshop, Atsaphone district)

Stakeholders also recommended expansion of the VHV network, suggesting recruiting new VHVs in all endemic areas for future malaria elimination activities, including malaria case management and surveillance.

*In the future, we will need more volunteers as we need to put more emphasis on surveillance and malaria case finding for elimination.* (Middle-level stakeholder, Nong District)

Regarding new volunteer selection criteria, community members wanted volunteers in middle age, of adequate education level, living locally and with experience of medical service provision. Female FGD participants requested at least one female volunteer for female patients

in the village, to avoid discomfort associated with consulting male volunteers about gender-sensitive health problems.

*Middle-aged volunteers are mature enough and still active. If they have adequate level of education and health care experience, they can easily learn in the training well and will be able to provide services for health problems in the village. We also want locally resident volunteers because we want to access the health care service 24/7 easily.* (Community member, participatory workshop, Nong District).

**Capacity building.**   To upgrade the skills and knowledge of volunteers to enable them to provide health care for a range of priority health problems, stakeholders suggested conducting refresher training sessions once in every three months. These would be organised by health officers from district health malaria units and district health offices under the supervision of the provincial health department.

Some stakeholders discussed developing a comprehensive manual to ensure understanding of and compliance with duties, policies, procedures, reference guidelines and other helpful information for volunteers. A middle-level stakeholder suggested that technical experts, stakeholders and implementing partners working in the malaria and public health sector should contribute to the creation of the new manual.

*Multi-level stakeholders and all implementing partners are encouraged to assist in designing a new manual which can provide overall structure, policies and guidance to volunteers throughout the volunteer process.* (Middle-level stakeholder)

**Malaria surveillance, monitoring and evaluation.**   Stakeholders noted that the National Malaria Strategic Plan includes a tailored 1-3-7 model [7] for the low burden provinces in the north of Lao PDR, whereby case notification is conducted within a day after diagnosis of a malaria case, individual case and foci investigation within three days and response within seven days. However, they reported that lack of human resource capacity in the national malaria program leads to inadequate case notification, foci investigation and response activity. Stakeholders wanted volunteers to take on the responsibility of case notification within one day of diagnosis by sending SMS messages to health centres and assisting in case-based surveillance and response activities to reduce the effects of human resources limitations in the National Malaria Program.

Stakeholders were aware of the importance of monitoring and evaluation visits for National Malaria Program. They were concerned about the irregularity of health centre staff visits because of insufficient time and human resources and poor road conditions. High-level stakeholders suggested that officers and field staff who speak local dialects should conduct quarterly visits to monitor volunteer performance with respect to service provision and data reporting and motivate the volunteers.

*In ethnic villages, all villagers including volunteers speak ethnic language only. Health staff who can speak ethnic language should conduct field visits to that community.* (High-level stakeholder)

In contrast, community members requested that high-level health stakeholders inspect the status of current malaria prevention activities personally and lead those activities in the community. They stated that VHVs become more active in malaria service provision immediately after supervision visits from high-level district health staff.

## Discussion

This study was conducted in the context of Lao's PDR malaria elimination program, with the aim of exploring the perspectives of health stakeholders, VHVs and community members on community-delivered models to assist in developing optimal community-delivered malaria elimination model for Lao PDR. Findings from interviews, FGDs and workshops indicated that malaria remains a significant health problem in the community. Therefore, interventions for malaria should be kept in the VHV model, while services for dengue, diarrhoea, influenza, TB and skin infections should be added, in response to the priorities of and requests from community members. Both community members and health stakeholders were in favour of developing an optimal community-delivered model that targets malaria elimination and covers services for the priority health problems. The proposed model includes at least three VHVs in a village–preferably at least one female VHV–who provide health services 24/7, strengthened and supported with quarterly refresher trainings and regular supervision visits conducted by district-level health staff.

Community members and leaders, as well as stakeholders, were aware that the malaria burden had declined dramatically in their area, but they still regarded malaria as important because of its huge impact on their communities in the past. This finding was consistent with other qualitative studies conducted in Lao PDR, which presented that community members described malaria as a major health problem [37,38]. Both community members and stakeholders accepted VHVs as front-line providers of malaria services and valued their work. This finding echoes a study conducted in Myanmar, which found that community members were willing to receive malaria testing from volunteers [39]. Collectively, these findings highlight that community members still want to receive malaria testing and treatment services from volunteers in these two GMS countries.

Robust epidemiological surveillance of transmission is fundamental to reducing the burden of malaria and achieving elimination [40,41]. In a strong surveillance system, malaria case-based data are recorded and reported in near-real time by health care providers so that elimination-specific activities can be executed promptly [42–44]. In Lao PDR, the malaria reporting mechanism is still reliant on monthly paper-based reporting by volunteers. Therefore, malaria elimination intervention such as case-based surveillance and response activity, the 1-3-7 model, is still challenging for malaria elimination because the very first step of elimination intervention, reporting of malaria cases, is delayed [7]. In other GMS countries, DHIS2 with real-time data is reported through mobile phone or tablet applications operated by health care providers, including volunteers [45,46]. However, in some areas of the GMS, mobile phone reporting is impractical because of insufficient mobile internet access in some remote areas of the country [47]. So, similar systems should be implemented initially in areas with sufficient mobile network coverage and internet access to upgrade VHVs reporting channel in Lao PDR, saving human resources and shortening reporting time before nationwide scale-up.

Community members and leaders were satisfied with the current malaria services provided by VHVs but requested additional primary care services for other priority health problems, namely dengue, diarrhoea, influenza, skin infections and TB. In the proposed model, malaria is still the primary disease to be addressed by volunteers. Routine volunteer interventions endorsed by Lao PDR's MOH will be continued and the new interventions added for malaria elimination may require advocacy with MOH to incorporate into the routine volunteer service package. Dengue is a climate-sensitive vector-borne disease, and Lao PDR experienced severe dengue outbreaks in 2013 and 2015 [48]. Dengue and malaria are similar epidemiologically, such as both diseases are single mode of transmission through a mosquito vector and increase in the rainy season. So, intervention strategies of both diseases such as personal protection and

larva control measures are overlapping and can be conducted together. Diarrhoea was recognized as one of the national notifiable diseases in 2004 and 11% of deaths in children under five are caused by diarrhoea [49]. Lao PDR remains a high TB burden country and with an estimated incidence of 155 all forms cases per 100,000 people [50]. Lao MOH aims to reduce diarrhoea and TB cases through Universal Health Coverage (UHC) including essential health, nutrition and TB services for all people in the community [51]. Many studies have pointed out that trained volunteers can be integrated effectively into primary health care settings for TB, diarrhoea, malaria and dengue [52–55]. The most challenging diseases in the proposed model to be approved by Lao PDR's MOH will be skin infection and treatment and referral of RDT-negative fever cases. But the proposed model is contributed by the MOH's stakeholder perspectives and opinions in model development. Moreover, in low and middle-income countries, the use of volunteers in the delivery of community-requested health care services appears promising [56]. An evidence-based, community-delivered, integrated malaria elimination model that responds to community priorities and health authorities' perspectives should be developed for malaria and the other five priority health problems in Lao PDR.

In the requested model, community members specified the volunteer selection criteria of being middle aged, being a local resident and having an adequate education level. They also requested to have at least three volunteers, including at least a female volunteer, per village. Along with the gender line, female volunteers predominantly engage in caring roles like health promotion and household visits, while male volunteers tend to take on more front-line roles in emergency response [57]; and female patients are more likely to seek primary health care services from female volunteers for gender sensitive health problems [58]. Moreover, studies from other settings have indicated that community choose volunteers with good character, honesty, diligence, the spirit of volunteerism, and prefer to select local residents rather than outsiders because of familiarity, cultural similarities and trustworthiness [59–62]. So, to enhance the health care seeking practice of community members, community requested that volunteer number and selection criteria should be considered in alignment with national malaria strategic plan of Lao PDR.

In their proposed model, community members requested health education sessions on preventive measures against all common diseases using easily digestible audiovisual materials and images. Although health literacy level in Savannakhet province is over 70% [24], the community members prefer pictorial and audio-visual health education approach because they perceived pictorial and audio-visual health messages as a potentially powerful element that can attract and communicate quickly [63,64]. Previous studies have demonstrated that providing health education through television, online and via loudspeakers improves disease prevention and care seeking practice [65–67]. To improve public awareness of malaria in the country, it is suggested that the national malaria program engages the community with malaria health messages through public media platforms via radio and television advertisements, short stories and songs using both first language and ethnic languages. Sociocultural determinants such as gender norms and low autonomy have been proposed as barriers that prevent women from accessing health care services [68,69], hence the value of loudspeaker-based health education sessions for women and girls who are busy with household chores. In addition, in Lao PDR, patriarchal cultural norms and values still exist [70] and it is suggested that household heads and community members who have decision making power of care should be engaged to improve awareness and health care seeking practice in the community. In these ways, awareness-raising activities for health care seeking practice targeted to common diseases, an integral part of disease prevention and control, could be conducted through the VHV network using the appropriate strategy and updated technology to create an enabling environment for all community members.

Stakeholders also highlighted the importance of regular monitoring and evaluation visits to ensure the quality of VHVs' care and to maintain their motivation. Monitoring and evaluation is a key part of the National Malaria Strategic Plan to achieve malaria elimination in Lao PDR by 2030 [8,20]. In the context of limited monitoring and supervision for malaria program, the district level health staff are yet to monitor and supervise VHVs for additional five disease control program if the existing model transformed into the community requested integrated model. In the proposed model, to improve monitoring and supervision in primary health care for both malaria and the additional five diseases in Lao PDR, national and international organisations should prepare and resource a monitoring and supervision plan integrated with the transition into a new VHV model.

All GMS countries (except Yunnan Province in China) are implementing the malaria volunteer model and are considering redesigning their current models in response to community demands and changing malaria epidemiology and national malaria elimination strategies [71]. In Lao PDR, VHVs provide services for both malaria and basic first aid care, while VMWs provide malaria services only in malaria endemic areas [8]. Due to the declining malaria burden and limited resources, VMWs and VHVs may need to be subsumed into a community-preferred integrated malaria elimination model in order to sustain their roles. Nevertheless, the integration of interventions into existing VMW model or restructuring of VHV model must be evidence-based. In addition to evidence, the proposed model would need to be backed by acceptability, fidelity, feasibility, cost-effectiveness issues, particularly funding, and political commitment.

Sustaining the community-delivered model in the wider national health system is important. The volunteer network needs to be part of the UHC scheme and primary health care system of a country. In addition to achieving malaria elimination in Lao PDR at 2025, the government has targeted achieving UHC by 2025 [51]. Three key dimensions of UHC are; essential health service coverage, financial risk protection, and equity in coverage [72]. If the community-preferred community-delivered health care model is developed and deployed, it will contribute to a higher level of essential health service coverage [73]. Lao PDR comprises 18 provinces with 8500 villages [24] where VHVs and VMWs are providing health services in the community. If they transform into the integrated volunteers, parallel financing of different silo volunteer models could be integrated as well. Hence, it may reduce the overall expenditure of different volunteer models in Lao PDR by improving financial risk protection and equity among rural populations.

Subclinical malaria, which can sustain a low level of remnant disease in the population [74–76], is another important issue for surveillance. It is critical to detect and treat both clinical and subclinical malaria and therefore it will be comprehensive national malaria surveillance system to achieve malaria elimination in a region [75,77,78]. No community members, leaders or volunteers discussed subclinical malaria, despite prompting. Consistent with the finding from another study in Lao PDR, more than half of the study population disagreed that a seemingly healthy person could have malaria parasites in their blood [37]. Similarly, community members and leaders were not aware of and did not discuss the unique challenges of and interventions for *P. falciparum* and *P. vivax* cases. Collectively, these findings highlight the importance of community health literacy with respect to crucial malaria concepts for the success of elimination programs in the GMS. Considering this, malaria elimination policies and interventions should be designed for a context in which people do not easily see a difference between *P. falciparum* and *P. vivax* malaria and do not understand the concept of sub-clinical malaria.

Sensitive diagnostics and interventions targeting sub-clinical malaria have the potential to be delivered by volunteers but were not raised by stakeholders in this study for inclusion in the

requested model. For example, volunteers may be trained to replace normal RDT with highly sensitive RDT [35], and to collect blood samples for malaria polymerase chain reaction testing, as previously successfully undertaken with village health volunteers in South East Myanmar [79]. While the use of more sensitive diagnostics was not raised during qualitative data collection, the integrated volunteers' role may be adapted to incorporate these functions. Similarly, a previous study in Lao PDR demonstrated that the success of a targeted malaria elimination campaign was underpinned by the contribution of volunteers who played an integral role in the implementation of MDA activities in the study villages, yet this approach was not discussed [80]. If MDA becomes policy, volunteers can participate in community engagement and advocacy, provide logistical support in drug administration and check treatment adherence of community members, as has been implemented in trials of MDA in other GMS countries [36,38,80–83]. MDA is a labor-intensive intervention, and its success requires community trust and engagement, particularly when the disease, such as malaria, is disappearing. When the community trust in the new proposed model develops, it should increase support for, and engagement of communities with, MDA as well.

In contrast to community members and leaders, the stakeholders were well aware of challenges for *P. vivax* elimination compared to *P. falciparum* elimination. Never-the-less, stakeholders did not recommend specific interventions for *P. vivax* elimination which may be due to possible introduction of new tools or regimens such as tafenoquine [84] or 7-day high-dose treatment regimen of primaquine [85] that are beyond the scope of development of the new proposed model. Instead, volunteer will undertake DOT for each and every dose of primaquine for 14 days ensuring compliance and radical cure of *P. vivax* adhering to the National Malaria Treatment Guidelines. If tafenoquine or 7-day high-dose primaquine regimen becomes policy in Lao PDR, the volunteer will continue DOT for these new drug regimens.

## Strengths and limitations

This study is the first attempt to integrate stakeholder and community voices together in the formulation of an evidence-based community-delivered healthcare model for Lao PDR. This study collected comprehensive views about model development from participants ranging from national-level stakeholders to grassroots community members. However, stakeholders from local and international non-governmental organisations implementing malaria programs in Lao PDR were not included in this study, meaning the findings may not necessarily represent their perspectives. Moreover, the stakeholders and community consultations were conducted in Savannakhet Province only, and their community health priorities may differ from those of other provinces. Although both VMW and VHV provide malaria services in Lao PDR, VMWs did not participate in this study. Therefore, the views of VHVs presented herein may not represent those of all volunteer categories (VHV and VMW) in Lao PDR.

## Conclusions and recommendations

Both community members and health stakeholders recommended the development of a community-delivered integrated malaria elimination model for Lao PDR. In addition to malaria elimination activities, interventions for the prevention, diagnosis and referral for treatment of cases of dengue, diarrhoea, influenza, skin infection and TB were recommended to be incorporated into the current VHV model. This research could form the basis of nationwide qualitative consultations for the development of an integrated community-delivered health care model, based on the perspectives of community and health stakeholders and in consideration of the available resources and current health policies in Lao PDR. Any proposed model should

be pilot tested to enable refinement before national scale-up and reviewed and revised periodically to reflect the changing epidemiology of diseases in Lao PDR.

## Supporting information

**S1 File. Services provided by village health volunteers (VHVs) and village malaria workers (VMWs).**
(DOCX)

**S2 File. Topic guides for focus group discussions and interviews.**
(DOCX)

## Acknowledgments

The authors would like to thank community members and leaders, village health volunteers and key health stakeholders from Savannakhet Provincial Health Department, district health departments and local health centres of the Ministry of Health, Lao PDR, who generously participated in the study. We sincerely acknowledge all the staff and volunteers from Lao Tropical and Public Health Institute who contributed in data collection and management, and administration. The authors gratefully acknowledge the contribution to this work of the Victorian Operational Infrastructure Support Program received by the Burnet Institute.

## Author Contributions

**Conceptualization:** Khampheng Phongluxa, Win Han Oo, Julia C. Cuts, Freya J. I. Fowkes.

**Data curation:** Khampheng Phongluxa, Julia C. Cuts.

**Formal analysis:** May Chan Oo.

**Funding acquisition:** Win Han Oo, Win Htike, Freya J. I. Fowkes.

**Investigation:** Khampheng Phongluxa, Sengchanh Kounnavong, Syda Xayyavong, Chanthaly Louangphaxay, Win Htike, Freya J. I. Fowkes.

**Methodology:** Khampheng Phongluxa, Win Han Oo, Sengchanh Kounnavong, Chanthaly Louangphaxay, Win Htike, Julia C. Cuts, Freya J. I. Fowkes.

**Project administration:** Khampheng Phongluxa, Win Han Oo, Sengchanh Kounnavong, Syda Xayyavong, Chanthaly Louangphaxay, Win Htike, Freya J. I. Fowkes.

**Resources:** Sengchanh Kounnavong, Syda Xayyavong, Chanthaly Louangphaxay.

**Software:** May Chan Oo, Julia C. Cuts, Kaung Myat Thu, Galau Naw Hkawng.

**Supervision:** Win Han Oo, Sengchanh Kounnavong, Syda Xayyavong.

**Validation:** May Chan Oo, Julia C. Cuts, Kaung Myat Thu, Galau Naw Hkawng.

**Visualization:** May Chan Oo.

**Writing – original draft:** May Chan Oo, Freya J. I. Fowkes.

**Writing – review & editing:** May Chan Oo, Khampheng Phongluxa, Win Han Oo, Win Htike, Julia C. Cuts, Kaung Myat Thu, Galau Naw Hkawng, Freya J. I. Fowkes.

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
