## [Decision Letter · Decision Letter 0]

22 Oct 2021

PONE-D-21-31478Perspectives of health and community stakeholders on community-delivered models of malaria elimination in Lao People’s Democratic Republic: A qualitative studyPLOS ONE

Dear Dr. Oo,

Thank you for submitting your manuscript to PLOS ONE. After careful consideration, we feel that it has merit but does not fully meet PLOS ONE’s publication criteria as it currently stands. Therefore, we invite you to submit a revised version of the manuscript that addresses the points raised during the review process.

Kindly address all comments raised by the reviewers, reviewer two has suggested to add a number of articles and I encourage you to only include articles directly relevant to the topic of your manuscript that will result in a significant improvement of your paper 

We look forward to receiving your revised manuscript.

Kind regards,

Benedikt Ley

Academic Editor

PLOS ONE

Journal Requirements:

"The funding was received from the National Health and Medical Research Council (Australian Centre for Research Excellence in Malaria Elimination (ACREME) (1134989) and Career Development Fellowship to FJIF (1166753); and its Independent Research Institute Infrastructure Support Scheme)"

"The funding was received from the National Health and Medical Research Council (Australian Centre for Research Excellence in Malaria Elimination (ACREME) (1134989) and Career Development Fellowship to FJIF (1166753); and its Independent Research Institute Infrastructure Support Scheme).  The funders had no role in study design, data collection and analysis, decision to publish, or preparation of the manuscript."

Additional Editor Comments:

Kindly address all comments from reviewer 1. Reviewer 2 has suggested to consider a number of articles and I encourage you to only include those articles that are relevant to the topic and will result in an improvement of your manuscript.

Reviewers' comments:

Reviewer's Responses to Questions

**Comments to the Author**

1. Is the manuscript technically sound, and do the data support the conclusions?

Reviewer #1: Partly

Reviewer #2: Yes

2. Has the statistical analysis been performed appropriately and rigorously? 

Reviewer #1: N/A

Reviewer #2: Yes

3. Have the authors made all data underlying the findings in their manuscript fully available?

Reviewer #1: No

Reviewer #2: Yes

4. Is the manuscript presented in an intelligible fashion and written in standard English?

Reviewer #1: Yes

Reviewer #2: Yes

5. Review Comments to the Author

Reviewer #1: Major general comments

1. The differentiation between village health volunteers (VHVs) and village malaria workers (VMWs) needs to be made clear at the beginning and usage of the language should be consistent throughout the manuscript. The fact that only VHVs were a part of the study and no VMWs should be made clear earlier than in the limitations section.

2. Nong district, one of the three districts included in the study, is said to not have VMWs, but it is a high burden district and according to current stratification should have VMWs.

3. It is my understanding that according to national guidelines, VHVs would not be testing and treating for malaria. They are only trained to refer. Please clarify.

4. Throughout the results section the authors present results by community members and stakeholders. In the methods section the sampling frame is stated to include MOH stakeholders, VHVs, community members and community leaders. It remains unclear if results from VHV are not included or if they are included in the community member group or in the stakeholder group.

5. For a lot of the findings it would be very interesting to contrast community views to VHVs view. The suggested model increases the work load and skill demand significantly. So the obvious question would be can VHV do this, how can they do this, do they want this?

6. There also seems to be a lack of detail on what type/kind of additional training would be required and how this might be achieved. I understand if this might be a next step in the process, but would encourage the authors to acknowledge this in the limitation section in more detail.

7. In conjunction with a more detailed discussion on training needs, the papers discussion would benefit from a more critical discussion around the feasibility of this new model. What level of care for those additional 5 diseases would be expected and might this just wishful thinking and a bit too far removed from what is realistically achievable?

8. The same applies to the health education (videos and films etc.) suggestions for VHV to add to their tasks. A more critical discussion on who would develop those, who would keep them up to date and how VHV can integrate this into the work routine would be useful. This is also relevant for the section on monitoring and supervision. With the addition of 5 more diseases the authors acknowledge the need for monitoring and supervision will increase, but as stated this is currently with only malaria included limited.

9. The authors discuss the fact that none of the community members or leaders mentioned different malaria species or the issue around subclinical malaria. This is really interesting and I would encourage the authors to add those findings into the result section. Was this the case for the VHV as well? Or did they mention those issues? If VHV also did not mention those, it would be again suitable to critically appraise the feasibility of adding 5 diseases to the portfolio.

Specific Comments

10. line 75-78: this sentence provides the rationale for your study but seems very speculative and without further evidence for the insufficiency of the current model. It would be good to support your statement and therefore the rationale for your study better.

11. Line 100-102. Although the authors reference later the question guides it would be helpful for the treader to articulate the themes that were explored in the more details.

12. Line 141-143. Same comment as for the previous comment. A short summary of the key themes as they relate to the questions guide/ agenda and/or results would be helpful at this stage of the reading.

13. Line 151. Here you state that the focus of the research were the opinions and knowledge whereas earlier you state that focus where perspectives and inputs. I understand that the first terms (opinions and perspectives) might be used interchangeable, but I feel there is quite a difference between knowledge and inputs.

Minor comments

14. Line 46/47: please add the year when the strategic plans aims to reach the mentioned targets

15. Line 45-46: “Malaria transmission is low and sporadic in the northern provinces but higher in the southern provinces, which account for 95% of all malaria cases in the country.” Which year is this percentage for? And what is the source?

16. Line 107: would clarify if interviews were undertaken in Lao language, or local dialects (which are not Lao).

17. Line 127 – Table 1 does not include central level as participants; however, previously in the methods section on line 102 central level is mentioned.

18. Line 286: would change the sentence to: “Some community members seek health care from traditional healers.” It’s quite an assumption that this is based on superstitious beliefs. Could be a lot of other more complex reasons.

19. Line 313: VHVs are effective resources. Would not say that people are a resource? That VHVs have effective resources?

20. Line 385: “Stakeholders wanted volunteers to the responsibility of case notification...” missing words here, -> “Stakeholders wanted volunteers to take on the responsibility of case notification…”

21. Line 439: mentions that VHVs should have tablets or mobile phones to upgrade reporting channels, but does not mention a significant limitation in terms of connectivity in many areas of malaria transmission. Would be more complete to at least mention connectivity challenges which are seen in other SEA countries as well.

22. Line 452-453: grammar/clarity changes” “…. the volunteer selection criteria of having middle age, being a local resident and having adequate education level.” Would change to “…the volunteer selection criteria of being middle aged, being a local resident and having an adequate education level.

23. Line 455-457: grammar/clarity changes: “Along with the gender line, female volunteers predominantly engage in caring roles like health promotion and household visits, while male volunteers tend to take on more front-line toles in emergency response (32); and female patients are more likely to seek primary care services from…”

24. Line 460-462: grammar/clarity changes: “So, to enhance the health care seeking practice of community members, the community requested that volunteer number and selection criteria should be considered in …”

25. In the results section the development of a manual for VHVs is mentioned; but the challenges in terms of literacy and capacity of VHVs is not mentioned. (Line 474-377). It is my understanding that for VMWs many do not read Lao language that well so having a manual in Lao might not be especially useful for VHVs. Might be worth noting, especially since in the following section on surveillance a high-level stakeholder quotation states: “In ethnic villages, all villagers including volunteers speak ethnic language only….” (Lines 397-399)

Reviewer #2: May Chan Oo and colleagues’ manuscript titled ‘perspectives of health and community stakeholders on community-delivered models of malaria elimination in Lao PDR-a qualitative study’ is a very well written manuscript. Manuscript is quite topical for GMS where sustaining and strengthening village health/malaria workers are in spotlight. I have comments and suggestions below, mostly to improve its scope, evidence base and discussion.

Overall

• It is essential to expand the roles and responsibilities of VMWs/VHWs to strengthen the primary/community health care. Authors have well discussed the topic, but at current, expansion of VHWs/VMWs is bottlenecked by evidence, and even more importantly funding and economic support to sustain these huge cadre of health workers. I urge authors to explore wider literature around how CHWs (VMWs/VHWs) are sustained in other low and middle income countries, both in Introduction and discuss them in Discussion. It is very important to marry the perspectives by stakeholders (from your study) to the sustainability issues (mostly funding and economy).

Specific

• In materials and methods, please add profile of these districts (or study setting/context) so that readers can understand how important it is to strengthen the network of CHWs.

• Semi-structured interviews: please explain what is the difference between key informant interviews and in-depth interviews, how were they different?

• Please explain briefly what is a qualitative descriptive approach.

• What do you mean by ‘…analysis was conducted at surface level’? please add a line about it.

In Results

• A lot of these findings are echoing with local evidence, evidence generated from some of these districts. There were studies around mass antimalarial administration in Nong where some of the elements you have touched on are extensively researched. I urge authors to explore all those literature and discuss them in ‘discussion’.

For example, malaria as a health concern, volunteer assignment and recruitment—piloted for MDA

Discussion

Overall, authors seem to overlook/omit local evidence, that are relevant for this study. Comparing your findings with local evidence substantiates your findings before you move on to discuss with the regional evidence. I urge authors to discuss with local evidence, a lot of these are generated from social science studies around MDA in Nong.

• Line 421-423: malaria as a health concern/priority problem. Please check local evidence

• Line 444-447: Please check local evidence on how village volunteers were trained and devolved responsibilities to effectively carry out a targeted malaria elimination research in Nong. This would already be piloted evidence to discuss with your findings.

• Line 458-460: The statement needs more discussion. I urge authors to explore reasons around it; it may allude to the fact that there is a preference for locals rather than outsiders, familiarity, cultural similarities, reassurance, trust, many of these factors may have played a role!

• Line 465-467: Discuss with local evidence, how pictorial messages were more comprehensive to community members. Again it links back to how low level of literacy in these areas demands audio-visual, pictorial messages.

• Line 469-472: Need to discuss more with local cultural context. While your statement is true, at the same time, household head has the most decision-making power, and there is high cohesion among community members, this demands engaging with households heads and community leaders for important health messages to improve the health seeking behaviour.

• Line 487-489: Yes, its under consideration, but may not be already implementing. Countries in GMS are still working to forge evidence around how they can integrate these CHWs into primary health care system, especially VMWs when their roles have been shrunk due to decline of malaria. There is a rush to generate evidence as well as known constraints such as funding and support to sustain these CHWs network. How much can we expand the roles and responsibilities of VMWs is currently undergoing formative research.

• Line 493-495: In addition to evidence, it would need to be backed by feasibility issues, particularly funding, and political commitment.

• Line 497-499: Sub-clinical malaria, it is an important topic to discuss. Again, I urge authors to explore literature from within Nong/Laos how these concepts have been explored in recent MDAs, how local community members perceive such a concept.

• Yes, and more discussion around species of malaria. Why do you think they may not have been aware of these two type of malaria?

Relevant literature for authors’ consideration:

PMID: 28914184

PMID: 29061133

PMID: 30533024

PMID: 29316932

PMID: 30390647

6. PLOS authors have the option to publish the peer review history of their article (what does this mean?). If published, this will include your full peer review and any attached files.

Reviewer #1: No

Reviewer #2: No

---

## [Author Response · Author response to Decision Letter 0]

29 Jan 2022

'Response to Editor'

We thank the Editor for constructive comments and guidelines. We ensure that our manuscript meets PLOS ONE's style requirement. We provide the funding statement in the cover letter as follow:

"The funding was received from the National Health and Medical Research Council (Australian Centre for Research Excellence in Malaria Elimination (ACREME) (1134989) and Career Development Fellowship to FJIF (1166753); and its Independent Research Institute Infrastructure Support Scheme). The funders had no role in study design, data collection and analysis, decision to publish, or preparation of the manuscript."

For the data availability, we would like to provide this information:

According to the National Ethics Committee for Health Research (NECHR), Lao PDR confidentiality must be maintained utmost. Therefore, the datasets generated and/or analysed during the current study are not publicly available as the study collected data from specific province, districts and villages in Lao PDR, and the information may be identifiable to particular individuals, risking a breach in confidentiality. For more information, please contact National Ethic Committee for Health Research; Dr. Phoutanong Thongmalayvong (nechr@gmail.com), Tel: +856-21-250670 ext 205, 209. 

Moreover, in the manuscript, we provide the figure of three studied districts in Savannakhet Province which is developed by the authors by using QGIS software (version 3.22.0) and Photoshop 2020. The figure is licensed under a CC BY 4.0.

'Response to Reviewers'

We thank the reviewers for their constructive comments which we have answered in turn below. We believe the changes substantially improves the manuscript for publication.

Reviewer #1: Major general comments

1. Comment: The differentiation between village health volunteers (VHVs) and village malaria workers (VMWs) needs to be made clear at the beginning and usage of the language should be consistent throughout the manuscript. The fact that only VHVs were a part of the study and no VMWs should be made clear earlier than in the limitations section.

Response: In the introduction, we have now clarified the differentiation between VHVs and VMW with the following edits (lines 61 – 76) as well as edited for consistency of language throughout:

“In Lao PDR, community-delivered models are used to implement key malaria interventions such as testing and treatment and bed nets, which have contributed to the declining malaria burden over the past decade. Throughout Lao PDR, there is a large community-based network of village health volunteers (VHVs) and village malaria workers (VMWs) providing services in hard-to-reach areas (Kounnavong, Gopinath, Hongvanthong, Khamkong, & Sichanthongthip, 2017; Ministry of Health, 2014). Malaria program-specific VMWs are assigned to provide malaria services only in remote high malaria transmission areas (annual parasite incidence > 10 per 1000 population) in Lao PDR (Bipin Adhikari et al., 2019; Center for Malariology, 2016; Phommanivong, Thongkham, Deyer, Rene, & Barennes, 2010) whereas VHVs provide the same malaria-specific services as well as other general integrated health services in hard-to-reach areas where there is no health centre (Bipin Adhikari et al., 2019; Center for Malariology, 2016; Phommanivong et al., 2010). The specific services provided by VHV and VMW are detailed in supporting file 1. Malaria services provided by both VMW and VHV include malaria testing and treatment, net distribution, education and surveillance, with VHV also providing general health services such as assisting health centre staff in outreach activities, providing health education and community health promotion services, offering basic first aid care, facilitating antenatal care and performing vital event surveillance (S1 File) (Kounnavong et al., 2017; Ministry of Health, 2014).VHV provide the vast majority of services across Lao PDR; in 2020, 92% of volunteers (13,089/14,227) were VHVs compared to 8% of volunteers (1,138/14,227) who were VMWs(P. a. E. Center for Malariology, 2020; Primary Health Unit, 2020). 

Supporting file 1. Services provided by village health volunteers (VHVs) and village malaria workers (VMWs)

Services VHV VMW

Malaria-specific health services 

Administer rapid diagnostic test √ √

Provide antimalarial treatment for uncomplicated malaria √ √

Refer complicated malaria and malaria with pregnancy patients to nearest health facility √ √

Organise the distribution of insecticide-treated bed nets √ √

Provide health education related to malaria prevention √ √

Serve as first-tier malaria surveillance units √ √

Compile and report village-level malaria data to health centres √ √

Other general health services 

Assist health centre staff in outreach activities √ 

Provide health education and community health promotion services √ 

Provide basic health care using first aid kit √ 

Facilitate antenatal care √ 

Vital event surveillance √ 

Patient referral √ 

We have now included in the materials and methods section (in addition to limitations section) that only VHVs participated in the study.

Lines 193 – 196 in material and methods sections now reads:

“Whilst both VMW and VHV work closely together to provide malaria services and were invited to participate in the three studied districts, only VHVs participated in this study because during the survey collection period, VMWs were not available to participate in this study.” 

2. Comment: Nong district, one of the three districts included in the study, is said to not have VMWs, but it is a high burden district and according to current stratification should have VMWs.

Response: 

Nong district is a high burden district (API > 10) and both VHVs and VMWs provide malaria services in the community. However, as highlighted in our previous response during the data collection period, only VHVs were available and participated in FGDs. 

3. It is my understanding that according to national guidelines, VHVs would not be testing and treating for malaria. They are only trained to refer. Please clarify.

Response: 

According to the national malaria treatment guideline of Lao PDR, VHVs are allowed to provide malaria rapid diagnostic test and antimalarial treatment to uncomplicated malaria patients. VHVs are encouraged to refer complicated malaria cases and malaria with pregnancy to nearest health facility. We have now included referral which was accidentally omitted in the original (Supporting information 1 (S1 file)). Please see addition in table above in response to comment 1.

4. Comment: Throughout the results section the authors present results by community members and stakeholders. In the methods section the sampling frame is stated to include MOH stakeholders, VHVs, community members and community leaders. It remains unclear if results from VHV are not included or if they are included in the community member group or in the stakeholder group.

Response: VHVs were included in the community member group. To clarify we have added the following in methods and results:

Lines 196 – 198 in material and method section now reads:

“Eighteen FGDs with VHVs and community members (combined but men and women separately; nine FGDs each) and nine participatory workshops with community leaders were undertaken (Table 1).”

Lines 246 – 249 in result section now reads:

“For the purpose of reporting, here provincial and district level health officers are referred to as high-level stakeholders and community-level health officers as middle-level stakeholders. VHVs and community members are collectively referred to as community members because VHVs also represented their views as members of the community.”

5. For a lot of the findings it would be very interesting to contrast community views to VHVs view. The suggested model increases the work load and skill demand significantly. So, the obvious question would be can VHV do this, how can they do this, do they want this?

Response: While we have highlighted contrasting community views, it is hard to dissect out why a contrasting finding (in amongst all contrasting findings) is specific to them being a VHV because VHVs views are representing their views as both a volunteer and member of the community.

It is expected that VHV workload and skill set would increase in the suggested model. Any evaluation of this, including the VHV perspectives on it, would be done in a formal evaluation of the proposed models effectiveness, acceptability, fidelity, feasibility, and cost-effectiveness during pilot implementation. This is out with the scope of the current study. 

6. There also seems to be a lack of detail on what type/kind of additional training would be required and how this might be achieved. I understand if this might be a next step in the process, but would encourage the authors to acknowledge this in the limitation section in more detail.

Response: The reviewer raises some interesting points about the types of training required for the volunteers and strategies to achieve the required trainings. In this study, the authors explored the perspectives of health and community stakeholders on current volunteer models to generate information to enable the development of a new model. However, the types of training required for the volunteers and strategies to achieve them were not included in the themes of the current question guides. It is out with the current scope of the study but should be explored when designing its implementation in the pilot testing phase.

7. In conjunction with a more detailed discussion on training needs, the papers discussion would benefit from a more critical discussion around the feasibility of this new model. What level of care for those additional 5 diseases would be expected and might this just wishful thinking and a bit too far removed from what is realistically achievable?

Response: We have illustrated that primary care level services for additional five diseases is requested from the health stakeholders and community members in result section (Line 381 to 383): 

“Stakeholders and community members agreed that a primary health care service package covering malaria and other health problems, provided by community-based volunteers, is needed.”

In discussion section, we have now discussed that feasibility of the implementation of new model, and primary care services for additional five diseases are expected from VHV. We have added the following in discussion section (Line 531 – 555):

Community members and leaders were satisfied with the current malaria services provided by VHVs but requested additional primary care services for other priority health problems, namely dengue, diarrhoea, influenza, skin infections and TB. In the proposed model, malaria is still the primary disease to be addressed by volunteers. Routine volunteer interventions endorsed by Lao PDR’s MOH will be continued and the new interventions added for malaria elimination may require advocacy with MOH to incorporate into the routine volunteer service package. Dengue is a climate-sensitive vector-borne disease, and Lao PDR experienced severe dengue outbreaks in 2013 and 2015 (Vannavong, Seidu, Stenström, Dada, & Overgaard, 2019). Dengue and malaria are similar epidemiologically, such as both diseases are single mode of transmission through a mosquito vector and increase in the rainy season. So, intervention strategies of both diseases such as personal protection and larva control measures are overlapping and can be conducted together. Diarrhoea was recognized as one of the national notifiable diseases in 2004 and 11% of deaths in children under five are caused by diarrhoea (Houatthongkham, 2020). Lao PDR remains a high TB burden country and with an estimated incidence of 155 all forms cases per 100,000 people (The World Bank, 2019). Lao MOH aims to reduce diarrhoea and TB cases through Universal Health Coverage including essential health, nutrition and TB services for all people in the community. (The Global Fund, 2020). Many studies have pointed out that trained volunteers can be integrated effectively into primary health care settings for TB, diarrhoea, malaria and dengue (Abongo, Ulo, & Karanja, 2020; Jennifer L. Brenner et al., 2017; Smith Paintain et al., 2014; Sommerfeld & Kroeger, 2012). The most challenging diseases in the proposed model to be approved by Lao PDR’s MOH will be skin infection and treatment and referral of RDT-negative fever cases. But the proposed model is contributed by the MOH’s stakeholder perspectives and opinions in model development. Moreover, in low and middle-income countries, the use of volunteers in the delivery of community-requested health care services appears promising (Woldie et al., 2018). An evidence-based, community-delivered, integrated malaria elimination model that responds to community priorities and health authorities’ perspectives should be developed for malaria and the other five priority health problems in Lao PDR.

Moreover, lines 599 to 602 in discussion section also mentioned about the monitoring and supervision in primary health care for both malaria and the additional five diseases.

“In the proposed model, to improve monitoring and supervision in primary health care for both malaria and the additional five diseases in Lao PDR, national and international organisations should prepare and resource a monitoring and supervision plan integrated with the transition into a new VHV model”.

8. The same applies to the health education (videos and films etc.) suggestions for VHV to add to their tasks. A more critical discussion on who would develop those, who would keep them up to date and how VHV can integrate this into the work routine would be useful. This is also relevant for the section on monitoring and supervision. With the addition of 5 more diseases the authors acknowledge the need for monitoring and supervision will increase, but as stated this is currently with only malaria included limited.

Response: The reviewer raises some interesting points about how the model will be implemented and sustained but this is out with the current scope of the study and best place in an assessment of pilot implementation of the new model. However, we have made some suggestions in terms of considerations for community engagement and monitoring and supervision plans:

In the discussion section, we have now added (Lines 578 – 581):

“To improve public awareness of malaria in the country, it is suggested that the national malaria program engages the community with malaria health messages through public media platforms via radio and television advertisements, short stories and songs using both first language and ethnic languages.”

Lines 599 to 602 in discussion section now reads:

“In the proposed model, to improve monitoring and supervision in primary health care for both malaria and the additional five diseases in Lao PDR, national and international organisations should prepare and resource a monitoring and supervision plan integrated with the transition into a new VHV model.”

9. The authors discuss the fact that none of the community members or leaders mentioned different malaria species or the issue around subclinical malaria. This is really interesting and I would encourage the authors to add those findings into the result section. Was this the case for the VHV as well? Or did they mention those issues? If VHV also did not mention those, it would be again suitable to critically appraise the feasibility of adding 5 diseases to the portfolio.

Response: We have now included in the result section that none of the community members or leaders mentioned different malaria species or issues of subclinical malaria. 

Lines 339 to 347 in result section now reads:

“Importantly, community members, leaders and VHVs did not understand the concept of subclinical malaria nor unique challenges of eliminating Plasmodium falciparum and Plasmodium vivax and did not discuss them in any FGDs or participatory workshop, despite prompts. In contrast to community members, leaders and volunteers, the stakeholders understood subclinical malaria however, they did not suggest any interventions for its diagnosis and treatment (such as the use of highly sensitive RDT (Vasquez et al., 2018) for diagnosis and MDA for elimination (von Seidlein et al., 2019)). Unlike community members and leaders, the stakeholders were well aware of challenges for P. vivax elimination compared to Plasmodium falciparum elimination. Unlike community members and leaders, the stakeholders concern the challenges of P. vivax elimination compared to P. falciparum elimination, but they did not recommend specific interventions for P. vivax elimination.”

Lines 531 – 555 in discussion section also mentioned about the feasibility of adding 5 diseases to the portfolio (see response to comment no. 7).

Specific Comments

10. line 75-78: this sentence provides the rationale for your study but seems very speculative and without further evidence for the insufficiency of the current model. It would be good to support your statement and therefore the rationale for your study better.

Response: In the introduction section, we have now included further evidence with the following edits (line 86 – 94): 

“In 2018, Distinct Surveillance and Response Guidelines were developed for case-based surveillance, focus investigations and responses in Lao PDR, whereby case notification is conducted within a day after diagnosis of a malaria case, individual case and foci investigation within three days and response within seven days (1-3-7 model). However, the 1-3-7 strategy has so far failed to be applied effectively without the help of VHVs and VMWs in Lao PDR (P. a. E. C. Center for Malariology, 2020). Furthermore, it is doubtful that current community-delivered models of VHVs and VMWs meet the requirements of the malaria elimination program in Lao PDR, such as real-time notification of malaria cases within 24 hours, and case-based surveillance and response activities.”

11. Line 100-102. Although the authors reference later the question guides it would be helpful for the treader to articulate the themes that were explored in the more details.

Response: In the material and methods section, we have now added the short summary of the key themes of semi-structured interview (Line 147 to 154):

The question guide of semi-structured interviews include six themes which are (i) current malaria situation and control measures in their respective working area (ii) views and perspectives on the current malaria VHV models (iii) policy and strategic barriers and enablers for malaria elimination (iv) operational barriers and enablers for malaria elimination using community delivered models (v) strategies to maintain the motivation and social role of VHVs in the community (vi) factors that need to be addressed during the transition from malaria control to elimination using community delivered models.

12. Line 141-143. Same comment as for the previous comment. A short summary of the key themes as they relate to the questions guide/ agenda and/or results would be helpful at this stage of the reading.

Response: In the material and methods section, we have now added the short summary of the key themes of FGD and participatory workshop (Line 183 to 191):

The guides of FGDs and participatory workshop include seven themes which are (i) current malaria situation and priority health problems in the respective community (ii) malaria control measures and available health services in the respective community (iii) views and perspectives on the current malaria VHV models (iv) policy and strategic barriers and enablers for effective malaria control and elimination in the community (v) available community supports for malaria control and elimination in their community (vi) strategies to maintain the motivation and social role of VHVs in the community (vii) culture, customs and norms of the ethnic communities that paly as barriers and enablers for effective malaria control and elimination in the community. 

13. Line 151. Here you state that the focus of the research were the opinions and knowledge whereas earlier you state that focus where perspectives and inputs. I understand that the first terms (opinions and perspectives) might be used interchangeable, but I feel there is quite a difference between knowledge and inputs.

Response: We will use consistently as “perspectives, knowledges and inputs” throughout the whole manuscript.

Minor comments

14. Line 46/47: please add the year when the strategic plans aims to reach the mentioned targets

Response:

Line 46-48 in introduction section now reads:

The national strategic plan for malaria control and elimination 2016-2020 focuses on burden reduction in five southern provinces and targets elimination in 13 northern provinces (Center for Malariology, 2016; P. a. E. C. Center for Malariology, 2020).

15. Line 45-46: “Malaria transmission is low and sporadic in the northern provinces but higher in the southern provinces, which account for 95% of all malaria cases in the country.” Which year is this percentage for? And what is the source?

Response: 

Line 44 to 46 in introduction section now reads:

In 2018, malaria transmission is low and sporadic in the northern provinces but higher in the southern provinces, which account for 95% of all malaria cases in the country (P. a. E. C. Center for Malariology, 2020).

16. Line 107: would clarify if interviews were undertaken in Lao language, or local dialects (which are not Lao).

Response:

Line 127 in material and methods section now reads:

“….were undertaken in Laotian”

17. Line 127 – Table 1 does not include central level as participants; however, previously in the methods section on line 102 central level is mentioned.

Response: 

Line 123-124 in material and methods section now reads:

….key MoH stakeholders (ranging from provincial to field office levels), VHVs, community members, and community leaders on community-delivered models

18. Line 286: would change the sentence to: “Some community members seek health care from traditional healers.” It’s quite an assumption that this is based on superstitious beliefs. Could be a lot of other more complex reasons.

Response: 

Line 368 in result section now reads:

“Some community members seek health care from traditional healers.”

19. Line 313: VHVs are effective resources. Would not say that people are a resource? That VHVs have effective resources?

Response: 

Line 395-396 in result section now reads:

Stakeholders also suggested that VHVs are effective at raising awareness about common health issues.

20. Line 385: “Stakeholders wanted volunteers to the responsibility of case notification...” missing words here, -> “Stakeholders wanted volunteers to take on the responsibility of case notification…”

Response: 

Line 469-470 in result section now reads:

“Stakeholders wanted volunteers to take on the responsibility of case notification …….”

21. Line 439: mentions that VHVs should have tablets or mobile phones to upgrade reporting channels, but does not mention a significant limitation in terms of connectivity in many areas of malaria transmission. Would be more complete to at least mention connectivity challenges which are seen in other SEA countries as well.

Response: In the result section, we have now mentioned the challenges of connectivity in other GMS countries as well. Line 525 - 529 in discussion section now reads:

“However, in some areas of the GMS, mobile phone reporting is impractical because of insufficient mobile internet access in some remote areas of the country (Win Han et al., 2021). So, similar systems should be implemented initially in sufficient mobile network coverage and internet access areas to upgrade VHVs reporting channel in Lao PDR, saving human resources and shortening reporting time before nationwide scale-up.”

22. Line 452-453: grammar/clarity changes” “…. the volunteer selection criteria of having middle age, being a local resident and having adequate education level.” Would change to “…the volunteer selection criteria of being middle aged, being a local resident and having an adequate education level.

Response: Line 557 - 558 in discussion section now reads:

In the requested model, community members specified the volunteer selection criteria of being middle aged, being a local resident and having an adequate education level.

23. Line 455-457: grammar/clarity changes: “Along with the gender line, female volunteers predominantly engage in caring roles like health promotion and household visits, while male volunteers tend to take on more front-line toles in emergency response (32); and female patients are more likely to seek primary care services from…”

Response: Line 559 - 563 in discussion section now reads:

Along with the gender line, female volunteers predominantly engage in caring roles like health promotion and household visits, while male volunteers tend to take on more front-line roles in emergency response (Cadesky, Baillie Smith, & Thomas, 2019); and female patients are more likely to seek primary health care services from female volunteers for gender sensitive health problems (Panday, Bissell, van Teijlingen, & Simkhada, 2017).

24. Line 460-462: grammar/clarity changes: “So, to enhance the health care seeking practice of community members, the community requested that volunteer number and selection criteria should be considered in …”

Response: Line 566 - 568 in discussion section now reads:

“So, to enhance the health care seeking practice of community members, community requested that volunteer number….”

25. In the results section the development of a manual for VHVs is mentioned; but the challenges in terms of literacy and capacity of VHVs is not mentioned. (Line 474-377). It is my understanding that for VMWs many do not read Lao language that well so having a manual in Lao might not be especially useful for VHVs. Might be worth noting, especially since in the following section on surveillance a high-level stakeholder quotation states: “In ethnic villages, all villagers including volunteers speak ethnic language only….” (Lines 397-399)

Response: VHVs in each village were selected according to criteria such as possessing good health, completion of at least a primary level of education, and willingness to work on a voluntary basis. In the manuscript, some stakeholders discussed developing a comprehensive manual for volunteers. But the result of this study needs to be evaluated, developing the new manual, publishing with both Lao official and ethnic languages, and distributing to all volunteers are beyond the scope of the current study. 

Reviewer #2: May Chan Oo and colleagues’ manuscript titled ‘perspectives of health and community stakeholders on community-delivered models of malaria elimination in Lao PDR-a qualitative study’ is a very well written manuscript. Manuscript is quite topical for GMS where sustaining and strengthening village health/malaria workers are in spotlight. I have comments and suggestions below, mostly to improve its scope, evidence base and discussion.

Overall

• It is essential to expand the roles and responsibilities of VMWs/VHWs to strengthen the primary/community health care. Authors have well discussed the topic, but at current, expansion of VHWs/VMWs is bottlenecked by evidence, and even more importantly funding and economic support to sustain these huge cadre of health workers. I urge authors to explore wider literature around how CHWs (VMWs/VHWs) are sustained in other low and middle income countries, both in Introduction and discuss them in Discussion. It is very important to marry the perspectives by stakeholders (from your study) to the sustainability issues (mostly funding and economy).

Response: Thank you very much for your comment.

In the introduction section, we have now added the literature of how CHWs are sustained in other low- and middle-income countries (Line 50 to 59):

In parallel with the switch from malaria control to elimination in GMS, the motivation and social role of malaria volunteers have quickly plummeted along with the decline of malaria burden (Nay Yi Yi Linn et al., 2018); therefore, the sustainability of the community-delivered model is impacted. Evidence has shown that in the context of lower malaria incidence, increasing the services that the community requested, and the malaria volunteers can provide, increases utilisation and enables more malaria testing (Win Han Oo, Gold, Moore, Agius, & Fowkes, 2019). For example, the integrated Community Case Management model that delivers case management in malaria, pneumonia, diarrhoea, malnutrition screening and nutrition counselling to children under five, has shown good acceptability in Africa (Muhumuza, Mutesi, Mutamba, Ampuriire, & Nangai, 2015; Nanyonjo, Nakirunda, Makumbi, Tomson, & Källander, 2012) and proved its effectiveness in reducing under five mortality and malaria burden in Africa (Abegunde et al., 2016; J. L. Brenner et al., 2011; Miller et al., 2014; Mubiru et al., 2015; Mukanga et al., 2012) as well as Myanmar (Moe Myint Oo, 2017). 

In the discussion section, we have now discussed how CHWs are sustained in other low- and middle-income countries (Line 616 to 627):

Sustaining the community-delivered model in the wider national health system is important. The volunteer network needs to be part of the UHC scheme and primary health care system of a country. In addition to achieving malaria elimination in Lao PDR at 2025, the government has targeted achieving UHC by 2025 (The Global Fund, 2020). Three key dimensions of UHC are; essential health service coverage, financial risk protection, and equity in coverage (World Health Organization & World Bank Group, 2014). If the community-preferred community-delivered health care model is developed and deployed, it will contribute to a higher level of essential health service coverage (Han et al., 2018). Lao PDR comprises 18 provinces with 8500 villages (Lao Statistics Bureau, 2016) where VHVs and VMWs are providing health services in the community. If they transform into the integrated volunteers, parallel financing of different silo volunteer models could be integrated as well. Hence, it may reduce the overall expenditure of different volunteer models in Lao PDR by improving financial risk protection and equity among rural populations. 

Specific

• In materials and methods, please add profile of these districts (or study setting/context) so that readers can understand how important it is to strengthen the network of CHWs.

Response: 

In material and methods section, we have now added the profile of three study districts (Line 107 to 112):

This study was conducted in Nong, Sepone and Atsaphone districts in Lao PDR’s southern Savannakhet Province, whose area are 1,833, 2,097, 1,482 km² respectively. The population density of Nong district is 18.75/km² while Sepone and Atsaphone are 31.06/km² and 43.69/km² respectively. The majority of the population are farmers and live in mountainous and forested areas. Among the three districts, Nong and Sepone are located close to the Vietnam border.

• Semi-structured interviews: please explain what is the difference between key informant interviews and in-depth interviews, how were they different?

Response: 

We have now briefly clarified the differences between KII and IDI, with references. Line 131 to 142 in material and methods section now reads:

Key informant interviews were used to collect data from high-level policymakers, decision-makers and managers from MoH, Lao PDR and, given their generalized knowledge of community-delivered models and malaria. Key informant interviews are qualitative in-depth interviews with people who know what is going on in the community. The purpose of key informant interviews is to collect information from a wide range of people—including community leaders, professionals, or residents — who have first-hand knowledge about the community (UCLA Center for Health Policy Research). In-depth interviews were conducted with middle-level managers, staff and technical advisors from MoH, Lao PDR and malaria IPs, to take advantage of their knowledge and personal experiences in managing, implementation and operating the community-delivered models and malaria interventions (Esterberg, 2002; Hansen, 2006). In-depth interviewing is a qualitative research technique that involves conducting intensive individual interviews with a small number of respondents to explore their perspectives on a particular idea, program, or situation (Carolyn Boyce & Neale, 2006).

• Please explain briefly what is a qualitative descriptive approach.

Response: 

A qualitative descriptive approach was applied to explore the experiences, opinions and knowledge of stakeholders, and volunteers, community members and leaders –the beneficiaries of the model. This approach enabled a comprehensive summary of the participants’ interactions and experiences with specific community-delivered models (Colorafi & Evans, 2016; Lambert & Lambert, 2012). 

• What do you mean by ‘…analysis was conducted at surface level’? please add a line about it.

Response: 

There are basically three levels of qualitative data analysis namely micro, surface and macro levels. The surface level analysis focus on themes, social patterns and understandings. It is a typical qualitative data analysis. However, it has a tendency to be overly descriptive and not analytical (Willis et al, 2007).

We have now briefly explained in line no. 224 to 226 in in material and methods section and it can now read:

“There are basically three levels of qualitative data analysis namely micro, surface and macro levels, and the surface level analysis focus in themes, social patterns and understandings.”

In Results

• A lot of these findings are echoing with local evidence, evidence generated from some of these districts. There were studies around mass antimalarial administration in Nong where some of the elements you have touched on are extensively researched. I urge authors to explore all those literature and discuss them in ‘discussion’.

For example, malaria as a health concern, volunteer assignment and recruitment—piloted for MDA

Response: Thank you very much for your comment.

Authors explored local evidence and literature of malaria as a health concern, volunteer assignment and MDA. Then, we discussed them in discussion section as follow: 

Malaria as a health concern (line no. 507 to 509)

Volunteer assignment (line no. 563 to 566) and 

MDA in Lao PDR (line no. 651 to 660)

For more details, please kindly see the below responses.

Discussion

Overall, authors seem to overlook/omit local evidence, that are relevant for this study. Comparing your findings with local evidence substantiates your findings before you move on to discuss with the regional evidence. I urge authors to discuss with local evidence, a lot of these are generated from social science studies around MDA in Nong.

• Line 421-423: malaria as a health concern/priority problem. Please check local evidence

Response: 

In the discussion section, we have now added the local evidence (Line 507 – 509):

This finding was consistent with other studies conducted in Lao PDR, which presented that community members described malaria as a major health problem (B. Adhikari, Phommasone, Kommarasy, et al., 2018; B. Adhikari, Phommasone, Pongvongsa, et al., 2018)

• Line 444-447: Please check local evidence on how village volunteers were trained and devolved responsibilities to effectively carry out a targeted malaria elimination research in Nong. This would already be piloted evidence to discuss with your findings.

Response: 

In the discussion section, we have now checked local evidence on how volunteers were trained to carry out mass drugs administration activities and discussed them in line 644 – 660:

Sensitive diagnostics and interventions targeting sub-clinical malaria have the potential to be delivered by volunteers but were not raised by stakeholders in this study for inclusion in the requested model. For example, volunteers may be trained to replace normal RDT with highly sensitive RDT (Vasquez et al., 2018), and to collect blood samples for malaria polymerase chain reaction testing, as previously successfully undertaken with village health volunteers in South East Myanmar (Win Han et al., 2018). While the use of more sensitive diagnostics was not raised during qualitative data collection, the integrated volunteers’ role may be adapted to incorporate these functions. Similarly, a previous study in Lao PDR demonstrated that the success of a targeted malaria elimination campaign was underpinned by the contribution of volunteers who played an integral role in the implementation of MDA activities in the study villages, yet this approach was not discussed (B. Adhikari et al., 2017). If MDA becomes policy, volunteers can participate in community engagement and advocacy, provide logistical support in drug administration and check treatment adherence of community members, as has been implemented in trials of MDA in other GMS countries (B. Adhikari et al., 2017; B. Adhikari, Phommasone, Kommarasy, et al., 2018; Bipin Adhikari et al., 2017; Lwin et al., 2015; Nguyen et al., 2017; von Seidlein et al., 2019). MDA is a labor-intensive intervention, and its success requires community trust and engagement, particularly when the disease, such as malaria, is disappearing. When the community trust in the new proposed model develops, it should increase support for, and engagement of communities with, MDA as well. 

• Line 458-460: The statement needs more discussion. I urge authors to explore reasons around it; it may allude to the fact that there is a preference for locals rather than outsiders, familiarity, cultural similarities, reassurance, trust, many of these factors may have played a role!

Response: 

In the discussion section, we have now added (Line 563 – 566):

Moreover, studies from other settings have indicated that community choose volunteers with good character, honesty, diligence, the spirit of volunteerism, and prefer to select local residents rather than outsiders because of familiarity, cultural similarities and trustworthiness (Chatio & Akweongo, 2017; Evelyn Sakeah et al., 2021; E. Sakeah et al., 2014; Vouking, Tamo, & Mbuagbaw, 2013).

• Line 465-467: Discuss with local evidence, how pictorial messages were more comprehensive to community members. Again it links back to how low level of literacy in these areas demands audio-visual, pictorial messages.

Response: 

In the discussion section, we have now discussed the level of literacy level and demands on pictorial and audio-visual health messages (Line 573 – 576):

Although health literacy level in Savannakhet province is over 70% (Lao Statistics Bureau, 2016), the community prefer pictorial and audio-visual health education approach because they perceived pictorial and audio-visual health messages as a potentially powerful element that can attract and communicate quickly (Sychareun, Hansana, Phengsavanh, Chaleunvong, & Tomson, 2015; Yoshida, Kobayashi, Sapkota, & Akkhavong, 2011).

• Line 469-472: Need to discuss more with local cultural context. While your statement is true, at the same time, household head has the most decision-making power, and there is high cohesion among community members, this demands engaging with households heads and community leaders for important health messages to improve the health seeking behaviour.

Response: 

In the discussion section, we have now added the local cultural context (Line 584 – 587):

In addition, in Lao PDR, patriarchal cultural norms and values still exist (CARE International in Lao PDR, 2018) and it is suggested that household heads and community members who have decision making power of care should be engaged to improve awareness and health care seeking practice in the community.

• Line 487-489: Yes, its under consideration, but may not be already implementing. Countries in GMS are still working to forge evidence around how they can integrate these CHWs into primary health care system, especially VMWs when their roles have been shrunk due to decline of malaria. There is a rush to generate evidence as well as known constraints such as funding and support to sustain these CHWs network. How much can we expand the roles and responsibilities of VMWs is currently undergoing formative research.

Response: This study is part of the ongoing formative research in this area in the GMS and in collaboration with national malaria control programs. The aim is to explore the perspectives of health and community stakeholders on current volunteer models to generate information to enable development of a new model. 

Authors would like to know how the roles and responsibilities of volunteers can be expanded and therefore authors are doing qualitative consultation to generate information to enable development of a new model in Myanmar and Lao PDR. In Myanmar, we are doing the field testing of new proposed model called “Community-delivered Integrated Malaria Elimination (CIME) model”. 

It is expected that there may be some constraints as time period, funding, technical and operational support to sustain the volunteer network in Lao PDR. Although the new model has been developed scientifically, it needs to be evaluated formally for effectiveness, acceptability, fidelity, feasibility, and cost-effectiveness of the new model compared to current VHV model. Then the new model will be refined based on the result of the evaluation result so that it could be further adapted and generalized in Lao PDR.

• Line 493-495: In addition to evidence, it would need to be backed by feasibility issues, particularly funding, and political commitment.

Response: Although the new model has been developed scientifically, it needs to be evaluated formally for effectiveness, acceptability, fidelity, feasibility, and cost-effectiveness of the new model compared to current VHV model. Then the new model will be refined based on the result of the evaluation result so that it could be further adapted and generalized in Lao PDR. 

However, we have added some suggestions in terms of consideration for integration of interventions into existing volunteer models. In the discussion section, we have now added (Line 612-614):

“In addition to evidence, the proposed model would need to be backed by acceptability, fidelity, feasibility, cost-effectiveness issues, particularly funding, and political commitment.”

• Line 497-499: Sub-clinical malaria, it is an important topic to discuss. Again, I urge authors to explore literature from within Nong/Laos how these concepts have been explored in recent MDAs, how local community members perceive such a concept.

Response: In discussion section, we have now discussed about sub-clinical malaria and MDA in Nong district.

In the discussion section, we have now added (Line 633 – 635):

Consistent with the finding from another study in Lao PDR, more than half of the study population disagreed that a seemingly healthy person could have malaria parasites in their blood (B. Adhikari, Phommasone, Pongvongsa, et al., 2018).

In the discussion section, we have now added (Line 644-660):

Sensitive diagnostics and interventions targeting sub-clinical malaria have the potential to be delivered by volunteers but were not raised by stakeholders in this study for inclusion in the requested model. For example, volunteers may be trained to replace normal RDT with highly sensitive RDT (Vasquez et al., 2018), and to collect blood samples for malaria polymerase chain reaction testing, as previously successfully undertaken with village health volunteers in South East Myanmar (Win Han et al., 2018). While the use of more sensitive diagnostics was not raised during qualitative data collection, the integrated volunteers’ role may be adapted to incorporate these functions. Similarly, a previous study in Lao PDR demonstrated that the success of a targeted malaria elimination campaign was underpinned by the contribution of volunteers who played an integral role in the implementation of MDA activities in the study villages, yet this approach was not discussed (B. Adhikari et al., 2017). If MDA becomes policy, volunteers can participate in community engagement and advocacy, provide logistical support in drug administration and check treatment adherence of community members, as has been implemented in trials of MDA in other GMS countries (B. Adhikari et al., 2017; B. Adhikari, Phommasone, Kommarasy, et al., 2018; Bipin Adhikari et al., 2017; Lwin et al., 2015; Nguyen et al., 2017; von Seidlein et al., 2019). MDA is a labor-intensive intervention, and its success requires community trust and engagement, particularly when the disease, such as malaria, is disappearing. When the community trust in the new proposed model develops, it should increase support for, and engagement of communities with, MDA as well. 

• Yes, and more discussion around species of malaria. Why do you think they may not have been aware of these two type of malaria?

Response: 

In result section, we have now added why community members, leaders and VHVs were not aware these two types of malaria (Line 339 to 342). It can read now as:

Importantly, community members, leaders and VHVs did not understand the concept of subclinical malaria nor unique challenges of eliminating Plasmodium falciparum and Plasmodium vivax and did not discuss them in any FGDs or participatory workshop, despite prompts.

In the discussion section, we have now added (Line 662 – 670):

In contrast to community members and leaders, the stakeholders were well aware of challenges for P. vivax elimination compared to P. falciparum elimination. Never-the-less, stakeholders did not recommend specific interventions for P. vivax elimination which may be due to possible introduction of new tools or regimens such as tafenoquine (Llanos-Cuentas et al., 2019) or 7-day high-dose treatment regimen of primaquine (Chu et al., 2019) that are beyond the scope of development of the new proposed model. Instead, volunteer will undertake DOT for each and every dose of primaquine for 14 days ensuring compliance and radical cure of Plasmodium vivax adhering to the National Malaria Treatment Guidelines. If tafenoquine or 7-day high-dose primaquine regimen becomes policy in Lao PDR, the volunteer will continue DOT for these new drug regimens. 

Relevant literature for authors’ consideration:

PMID: 28914184

PMID: 29061133

PMID: 30533024

PMID: 29316932

PMID: 30390647

Response: Thank you very much for your suggestion. Authors have discussed and cited the recommended manuscripts in this study.

Reference:

Abegunde, D., Orobaton, N., Bassi, A., Oguntunde, O., Bamidele, M., Abdulkrim, M., & Nwizugbe, E. (2016). The Impact of Integrated Community Case Management of Childhood Diseases Interventions to Prevent Malaria Fever in Children Less than Five Years Old in Bauchi State of Nigeria. PLoS One, 11(2), e0148586. doi:10.1371/journal.pone.0148586

Abongo, T., Ulo, B., & Karanja, S. (2020). Community health volunteers’ contribution to tuberculosis patients notified to National Tuberculosis program through contact investigation in Kenya. BMC Public Health, 20(1), 1184. doi:10.1186/s12889-020-09271-7

Adhikari, B., Pell, C., Phommasone, K., Soundala, X., Kommarasy, P., Pongvongsa, T., . . . Cheah, P. Y. (2017). Elements of effective community engagement: lessons from a targeted malaria elimination study in Lao PDR (Laos). Glob Health Action, 10(1), 1366136. doi:10.1080/16549716.2017.1366136

Adhikari, B., Phommasone, K., Kommarasy, P., Soundala, X., Souvanthong, P., Pongvongsa, T., . . . Pell, C. (2018). Why do people participate in mass anti-malarial administration? Findings from a qualitative study in Nong District, Savannakhet Province, Lao PDR (Laos). Malar J, 17(1), 15. doi:10.1186/s12936-017-2158-4

Adhikari, B., Phommasone, K., Pongvongsa, T., Kommarasy, P., Soundala, X., Henriques, G., . . . Mayxay, M. (2017). Factors associated with population coverage of targeted malaria elimination (TME) in southern Savannakhet Province, Lao PDR. Malar J, 16(1), 424. doi:10.1186/s12936-017-2070-y

Adhikari, B., Phommasone, K., Pongvongsa, T., Koummarasy, P., Soundala, X., Henriques, G., . . . Mayxay, M. (2019). Treatment-seeking behaviour for febrile illnesses and its implications for malaria control and elimination in Savannakhet Province, Lao PDR (Laos): a mixed method study. BMC health services research, 19(1), 252-252. doi:10.1186/s12913-019-4070-9

Adhikari, B., Phommasone, K., Pongvongsa, T., Soundala, X., Koummarasy, P., Henriques, G., . . . Pell, C. (2018). Perceptions of asymptomatic malaria infection and their implications for malaria control and elimination in Laos. PLoS One, 13(12), e0208912. doi:10.1371/journal.pone.0208912

Brenner, J. L., Barigye, C., Maling, S., Kabakyenga, J., Nettel-Aguirre, A., Buchner, D., . . . Singhal, N. (2017). Where there is no doctor: can volunteer community health workers in rural Uganda provide integrated community case management? African health sciences, 17(1), 237-246. doi:10.4314/ahs.v17i1.29

Brenner, J. L., Kabakyenga, J., Kyomuhangi, T., Wotton, K. A., Pim, C., Ntaro, M., . . . Singhal, N. (2011). Can volunteer community health workers decrease child morbidity and mortality in southwestern Uganda? An impact evaluation. PLoS One, 6(12), e27997. doi:10.1371/journal.pone.0027997

Cadesky, J., Baillie Smith, M., & Thomas, N. (2019). The gendered experiences of local volunteers in conflicts and emergencies. Gender & Development, 27(2), 371-388. doi:10.1080/13552074.2019.1615286

CARE International in Lao PDR. (2018). Water for Women Gender, Equality and Social Inclusion Analysis. Retrieved from 

Carolyn Boyce, & Neale, P. (2006). Conducting In-depth Interviews: A Guide for Designing and Conducting In-Depth Interviews for Evaluation Input. In Pathfinder International tool series; 

Monitoring and Evaluation – 2 (pp. 16): Pathfinder International.

Center for Malariology, P. a. E. (2016). National Strategic Plan for Malaria Control and Elimination 2016-2020. Vientiane Capital, Lao PDR: Ministry of Health Retrieved from https://www.google.com/url?sa=t&rct=j&q=&esrc=s&source=web&cd=3&cad=rja&uact=8&ved=2ahUKEwisydyEm83kAhVHgI8KHc4NCjkQFjACegQIARAC&url=https%3A%2F%2Fwww2.malariafreemekong.org%2Fwp-content%2Fuploads%2F2019%2F03%2FLaos_Malaria-NSP.pdf&usg=AOvVaw2C5RKICSll473gF4OSkOeG

Center for Malariology, P. a. E. (2020). Lao PDR Malaria Program Review 2019. Lao PDR: Ministry of Health.

Center for Malariology, P. a. E. C. (2020). Lao PDR Malaria Program Review 2019. Lao PDR: Ministry of Health.

Chatio, S., & Akweongo, P. (2017). Retention and sustainability of community-based health volunteers' activities: A qualitative study in rural Northern Ghana. PLoS One, 12(3), e0174002. doi:10.1371/journal.pone.0174002

Chu, C. S., Phyo, A. P., Turner, C., Win, H. H., Poe, N. P., Yotyingaphiram, W., . . . White, N. J. (2019). Chloroquine Versus Dihydroartemisinin-Piperaquine With Standard High-dose Primaquine Given Either for 7 Days or 14 Days in Plasmodium vivax Malaria. Clin Infect Dis, 68(8), 1311-1319. doi:10.1093/cid/ciy735

Colorafi, K. J., & Evans, B. (2016). Qualitative Descriptive Methods in Health Science Research. Herd, 9(4), 16-25. doi:10.1177/1937586715614171

Esterberg, K. G. (2002). Qualitative methods in social research. Boston : McGraw-Hill, c2002.

Han, S. M., Rahman, M. M., Rahman, M. S., Swe, K. T., Palmer, M., Sakamoto, H., . . . Shibuya, K. (2018). Progress towards universal health coverage in Myanmar: a national and subnational assessment. The Lancet Global Health, 6(9), e989-e997. doi:10.1016/S2214-109X(18)30318-8

Hansen, E. C. (2006). Successful qualitative health research: A practical introduction (First ed.). 83 Alenxander Street, Crows Nest NSW 2065, Australia Allen & Unwin.

Houatthongkham, S. (2020). Etiologic agents of diarrhea in Vientiane Capital, Lao People's Democratic Republic. International Journal of Infectious Diseases, 101, 344. doi:https://doi.org/10.1016/j.ijid.2020.09.905

Kounnavong, S., Gopinath, D., Hongvanthong, B., Khamkong, C., & Sichanthongthip, O. (2017). Malaria elimination in Lao PDR: the challenges associated with population mobility. Infectious diseases of poverty, 6(1), 81-81. doi:10.1186/s40249-017-0283-5

Lambert, V. A., & Lambert, C. E. (2012). Qualitative Descriptive Research: An Acceptable Design. Pacific Rim Int J Nurs Res, 16, 2. Retrieved from https://www.tci-thaijo.org/index.php/PRIJNR/article/download/5805/5064

Lao Statistics Bureau. (2016). Results of population and housing census 2015. Vientiane: Lao Statistics Bureau. 

Llanos-Cuentas, A., Lacerda, M. V. G., Hien, T. T., Vélez, I. D., Namaik-Larp, C., Chu, C. S., . . . Green, J. A. (2019). Tafenoquine versus Primaquine to Prevent Relapse of Plasmodium vivax Malaria. N Engl J Med, 380(3), 229-241. doi:10.1056/NEJMoa1802537

Lwin, K. M., Imwong, M., Suangkanarat, P., Jeeyapant, A., Vihokhern, B., Wongsaen, K., . . . Nosten, F. (2015). Elimination of Plasmodium falciparum in an area of multi-drug resistance. Malar J, 14(1), 319. doi:10.1186/s12936-015-0838-5

Miller, N. P., Amouzou, A., Tafesse, M., Hazel, E., Legesse, H., Degefie, T., . . . Bryce, J. (2014). Integrated community case management of childhood illness in Ethiopia: implementation strength and quality of care. Am J Trop Med Hyg, 91(2), 424-434. doi:10.4269/ajtmh.13-0751

Ministry of Health. (2014). Lao People’s Democratic Republic Health System Review: Health Systems in Transition (Vol. 4): World Health Organization.

Moe Myint Oo. (2017). Integrated Community Case Management Dissemination and Consultation Meeting. Research dissemination meeting. Malaria Consortium. Meeting Hall, Disease Control Office, MoHS, Nay Pyi Taw. 

Mubiru, D., Byabasheija, R., Bwanika, J. B., Meier, J. E., Magumba, G., Kaggwa, F. M., . . . Diaz, T. (2015). Evaluation of Integrated Community Case Management in Eight Districts of Central Uganda. PLoS One, 10(8), e0134767. doi:10.1371/journal.pone.0134767

Muhumuza, G., Mutesi, C., Mutamba, F., Ampuriire, P., & Nangai, C. (2015). Acceptability and Utilization of Community Health Workers after the Adoption of the Integrated Community Case Management Policy in Kabarole District in Uganda. Health Syst Policy Res, 2(1). 

Mukanga, D., Tibenderana, J. K., Peterson, S., Pariyo, G. W., Kiguli, J., Waiswa, P., . . . Kallander, K. (2012). Access, acceptability and utilization of community health workers using diagnostics for case management of fever in Ugandan children: a cross-sectional study. Malar J, 11, 121. doi:10.1186/1475-2875-11-121

Nanyonjo, A., Nakirunda, M., Makumbi, F., Tomson, G., & Källander, K. (2012). Community acceptability and adoption of integrated community case management in Uganda. American Journal of Tropical Medicine and Hygiene, 87(SUPPL.5), 97-104. 

Nay Yi Yi Linn, Tripathy, J. P., Maung, T. M., Saw, K. K., Maw, L. Y. W., Thapa, B., . . . Thi, A. (2018). How are the village health volunteers deliver malaria testing and treatment services and what are the challenges they are facing? A mixed methods study in Myanmar. Tropical medicine and health, 46, 28-28. doi:10.1186/s41182-018-0110-0

Nguyen, T.-N., Thu, P. N. H., Hung, N. T., Son, D. H., Tien, N. T., Van Dung, N., . . . Hien, T. T. (2017). Community perceptions of targeted anti-malarial mass drug administrations in two provinces in Vietnam: a quantitative survey. Malar J, 16(1), 17. doi:10.1186/s12936-016-1662-2

Panday, S., Bissell, P., van Teijlingen, E., & Simkhada, P. (2017). The contribution of female community health volunteers (FCHVs) to maternity care in Nepal: a qualitative study. BMC health services research, 17(1), 623-623. doi:10.1186/s12913-017-2567-7

Phommanivong, V., Thongkham, K., Deyer, G., Rene, J. P., & Barennes, H. (2010). An assessment of early diagnosis and treatment of malaria by village health volunteers in the Lao PDR. Malar J, 9, 347-347. doi:10.1186/1475-2875-9-347

Primary Health Unit, D. o. H. a. H. p. (2020). Report on contribution of Village Health Volunteer in Lao PDR. Retrieved from Lao PDR: 

Sakeah, E., Aborigo, R. A., Debpuur, C., Nonterah, E. A., Oduro, A. R., & Awoonor-Williams, J. K. (2021). Assessing selection procedures and roles of Community Health Volunteers and Community Health Management Committees in Ghana’s Community-based Health Planning and Services program. PLoS One, 16(5), e0249332. doi:10.1371/journal.pone.0249332

Sakeah, E., McCloskey, L., Bernstein, J., Yeboah-Antwi, K., Mills, S., & Doctor, H. V. (2014). Is there any role for community involvement in the community-based health planning and services skilled delivery program in rural Ghana? BMC health services research, 14, 340. doi:10.1186/1472-6963-14-340

Smith Paintain, L., Willey, B., Kedenge, S., Sharkey, A., Kim, J., Buj, V., . . . Ngongo, N. (2014). Community Health Workers and Stand-Alone or Integrated Case Management of Malaria: A Systematic Literature Review. The American Society of Tropical Medicine and Hygiene, 91(3), 461-470. doi:10.4269/ajtmh.14-0094

Sommerfeld, J., & Kroeger, A. (2012). Eco-bio-social research on dengue in Asia: a multicountry study on ecosystem and community-based approaches for the control of dengue vectors in urban and peri-urban Asia. Pathog Glob Health, 106(8), 428-435. doi:10.1179/2047773212y.0000000055

Sychareun, V., Hansana, V., Phengsavanh, A., Chaleunvong, K., & Tomson, T. (2015). Perceptions and acceptability of pictorial health warning labels vs text only - a cross-sectional study in Lao PDR. BMC Public Health, 15(1), 1094. doi:10.1186/s12889-015-2415-9

The Global Fund. (2020). Lao PDR aims to achieve universal health coverage with new Global Fund, Government of Australia and World Bank investment. Retrieved from https://www.theglobalfund.org/en/news/2020-11-18-lao-pdr-universal-health-coverage-global-fund-government-of-australia-and-world-bank-investment/

The World Bank. (2019). Incidence of tuberculosis in Lao PDR. Retrieved from https://data.worldbank.org/indicator/SH.TBS.INCD?locations=LA. Retrieved 1 Dec 2021 https://data.worldbank.org/indicator/SH.TBS.INCD?locations=LA

UCLA Center for Health Policy Research. Key Informant Interviews. In (pp. 10). Online: University of California, Los Angeles.

Vannavong, N., Seidu, R., Stenström, T. A., Dada, N., & Overgaard, H. J. (2019). Dengue-like illness surveillance: a two-year longitudinal survey in suburban and rural communities in the Lao People's Democratic Republic and in Thailand. Western Pacific Surveillance and Response Journal : WPSAR, 10, 15 - 24. 

Vasquez, A. M., Medina, A. C., Tobon-Castano, A., Posada, M., Velez, G. J., Campillo, A., . . . Ding, X. (2018). Performance of a highly sensitive rapid diagnostic test (HS-RDT) for detecting malaria in peripheral and placental blood samples from pregnant women in Colombia. PLoS One, 13(8), e0201769. doi:10.1371/journal.pone.0201769

von Seidlein, L., Peto, T. J., Landier, J., Nguyen, T.-N., Tripura, R., Phommasone, K., . . . White, N. J. (2019). The impact of targeted malaria elimination with mass drug administrations on falciparum malaria in Southeast Asia: A cluster randomised trial. PLOS Medicine, 16(2), e1002745. doi:10.1371/journal.pmed.1002745

Vouking, M. Z., Tamo, V. C., & Mbuagbaw, L. (2013). The impact of community health workers (CHWs) on Buruli ulcer in sub-Saharan Africa: a systematic review. Pan Afr Med J, 15, 19. doi:10.11604/pamj.2013.15.19.1991

Win Han, O., Cutts, J. C., Agius, P. A., Kyaw Zayar, A., Poe Poe, A., Aung, T., . . . Fowkes, F. J. I. (2018). Effectiveness of repellent delivered through village health volunteers on malaria incidence in villages in South-East Myanmar: a stepped-wedge cluster-randomised controlled trial protocol. BMC Infect Dis, 18(1), 663. doi:10.1186/s12879-018-3566-y

Win Han, O., Kaung Myat, T., Cutts, J. C., Win, H., Kyawt Mon, W., May Chan, O., . . . Fowkes, F. J. I. (2021). Sustainability of a mobile phone application-based data reporting system in Myanmar’s malaria elimination program: a qualitative study. BMC Medical Informatics and Decision Making, 21(1), 285. doi:10.1186/s12911-021-01646-z

Win Han Oo, Gold, L., Moore, K., Agius, P. A., & Fowkes, F. J. I. (2019). The impact of community-delivered models of malaria control and elimination: a systematic review. Malar J, 18(1), 269. doi:10.1186/s12936-019-2900-1

Woldie, M., Feyissa, G. T., Admasu, B., Hassen, K., Mitchell, K., Mayhew, S., . . . Balabanova, D. (2018). Community health volunteers could help improve access to and use of essential health services by communities in LMICs: an umbrella review. Health Policy and Planning, 33(10), 1128-1143. doi:10.1093/heapol/czy094

World Health Organization, & World Bank Group. (2014). Monitoring progress towards universal health coverage at country and global levels(pp. 14). Retrieved from https://www.google.com/url?sa=t&rct=j&q=&esrc=s&source=web&cd=1&cad=rja&uact=8&ved=2ahUKEwjLta7ppIjiAhVf6nMBHdyaD70QFjAAegQIAhAC&url=https%3A%2F%2Fwww.who.int%2Firis%2Fbitstream%2F10665%2F112824%2F1%2FWHO_HIS_HIA_14&usg=AOvVaw0WlHk5jAydMZKosDKeF20s

Yoshida, I., Kobayashi, T., Sapkota, S., & Akkhavong, K. (2011). Evaluating educational media using traditional folk songs (‘lam’) in Laos: a health message combined with oral tradition. Health Promotion International, 27(1), 52-62. doi:10.1093/heapro/dar086

---

## [Decision Letter · Decision Letter 1]

10 Feb 2022

Perspectives of health and community stakeholders on community-delivered models of malaria elimination in Lao People’s Democratic Republic: A qualitative study

PONE-D-21-31478R1

Dear Dr. Oo,

We’re pleased to inform you that your manuscript has been judged scientifically suitable for publication and will be formally accepted for publication once it meets all outstanding technical requirements.

Kind regards,

Benedikt Ley, PhD

Academic Editor

PLOS ONE

Additional Editor Comments (optional):

Reviewers' comments:

Reviewer's Responses to Questions

**Comments to the Author**

1. If the authors have adequately addressed your comments raised in a previous round of review and you feel that this manuscript is now acceptable for publication, you may indicate that here to bypass the “Comments to the Author” section, enter your conflict of interest statement in the “Confidential to Editor” section, and submit your "Accept" recommendation.

Reviewer #1: All comments have been addressed

Reviewer #2: All comments have been addressed

2. Is the manuscript technically sound, and do the data support the conclusions?

Reviewer #1: Yes

Reviewer #2: Yes

3. Has the statistical analysis been performed appropriately and rigorously? 

Reviewer #1: N/A

Reviewer #2: N/A

4. Have the authors made all data underlying the findings in their manuscript fully available?

Reviewer #1: No

Reviewer #2: Yes

5. Is the manuscript presented in an intelligible fashion and written in standard English?

Reviewer #1: Yes

Reviewer #2: Yes

6. Review Comments to the Author

Reviewer #1: All comments have been addressed. xxxxxxxxxxxxxxxxxxxxxxxxxxxxxxxxxxxxxxxxxxxxxxxxxxxxxxxxxxxxxxxxxxxxxxxxxxxx

Reviewer #2: (No Response)

7. PLOS authors have the option to publish the peer review history of their article (what does this mean?). If published, this will include your full peer review and any attached files.

Reviewer #1: No

Reviewer #2: No

---

## [Editor Report · Acceptance letter]

1 Mar 2022

PONE-D-21-31478R1 

Perspectives of health and community stakeholders on community-delivered models of malaria elimination in Lao People’s Democratic Republic: A qualitative study

Dear Dr. Oo:

I'm pleased to inform you that your manuscript has been deemed suitable for publication in PLOS ONE. Congratulations! Your manuscript is now with our production department. 

Kind regards, 

on behalf of

Dr Benedikt Ley 

Academic Editor

PLOS ONE